# Tau follows principal axes of functional and structural brain organization in Alzheimer's disease

Julie Ottoy [1], Min Su Kang [1], Jazlynn Xiu Min Tan[1], Lyndon Boone[1], Reinder Vos de Wael[2], Bo-yong Park [3,4], Gleb Bezgin[5,6], Firoza Z. Lussier[5,7], Tharick A. Pascoal[7], Nesrine Rahmouni [5], Jenna Stevenson[5], Jaime Fernandez Arias[5], Joseph Therriault [5], Seok-Jun Hong [8], Bojana Stefanovic[1,9,10], JoAnne McLaurin[1,11,12], Jean-Paul Soucy[2], Serge Gauthier[5], Boris C. Bernhardt [2], Sandra E. Black [1,13], Pedro Rosa-Neto [2,5] & Maged Goubran [1,9,10] ✉

Alzheimer's disease (AD) is a brain network disorder where pathological proteins accumulate through networks and drive cognitive decline. Yet, the role of network connectivity in facilitating this accumulation remains unclear. Using in-vivo multimodal imaging, we show that the distribution of tau and reactive microglia in humans follows spatial patterns of connectivity variation, the so-called *gradients* of brain organization. Notably, less distinct connectivity patterns ("gradient contraction") are associated with cognitive decline in regions with greater tau, suggesting an interaction between reduced network differentiation and tau on cognition. Furthermore, by modeling tau in subject-specific gradient space, we demonstrate that tau accumulation in the fronto-parietal and temporo-occipital cortices is associated with greater baseline tau within their functionally and structurally connected hubs, respectively. Our work unveils a role for both functional and structural brain organization in pathology accumulation in AD, and supports subject-specific gradient space as a promising tool to map disease progression.

Amyloid-β (Aβ) plaques and neurofibrillary tau tangles are key pathologic substrates for the spectrum disorder called Alzheimer's disease (AD), the most common cause of dementia and one of the most feared consequences of aging[1]. Aβ and tau progressively aggregate in the AD brain, driving downstream synaptic and neuronal loss and cognitive impairment[2,3]. The severity of AD-related tauopathy (assessed by imaging or biofluid measurements) outperforms that of Aβ in predicting future neuronal loss and domain-specific

[1]Sunnybrook Research Institute, University of Toronto, Toronto, ON, Canada. [2]McConnell Brain Imaging Centre, Montreal Neurological Institute and Hospital, McGill University, Montreal, Quebec, Canada. [3]Department of Data Science, Inha University, Incheon, Republic of Korea. [4]Center for Neuroscience Imaging Research, Institute for Basic Science, Suwon, Republic of Korea. [5]Translational Neuroimaging laboratory, McGill Centre for Studies in Aging, McGill University, Montreal, QC, Canada. [6]Neuroinformatics for Personalized Medicine lab, Montreal Neurological Institute, McGill University, Montréal, QC, Canada. [7]Department of Psychiatry, University of Pittsburgh, Pittsburgh, PA, USA. [8]Department of Biomedical Engineering, Sungkyunkwan University, Suwon, Republic of Korea. [9]Department of Medical Biophysics, University of Toronto, Toronto, ON, Canada. [10]Physical Sciences Platform, Sunnybrook Research Institute, University of Toronto, Toronto, ON, Canada. [11]Department of Laboratory Medicine and Pathobiology, University of Toronto, Toronto, ON, Canada. [12]Biological Sciences Platform, Sunnybrook Research Institute, University of Toronto, Toronto, ON, Canada. [13]Department of Medicine (Division of Neurology), University of Toronto, Toronto, ON, Canada. ✉e-mail: maged.goubran@utoronto.ca

cognitive decline in individual patients[4]. As such, limiting the spread of tau is a critical target for treatment[5]. It is thus imperative to gain a comprehensive understanding of how tau accumulates throughout the brain.

Traditionally, tau is thought to follow the hierarchical staging scheme based on neuropathological observations by Braak and Braak[6,7]. Tau tangles start in the (trans-)entorhinal cortex and locus coeruleus, then arise in limbic areas and the neocortex, triggering pro-inflammatory responses[8]. While in-vivo positron emission tomography (PET) imaging of tau and neuroinflammation has replicated Braak staging at the group level[9,10], this staging scheme is less sensitive to heterogeneity at the individual subject level[11,12]; posing challenges in the development of personalized prediction models of tau progression and patient stratification.

One promising approach is to incorporate brain networks, also known as the connectome, in individual models of tau accumulation[13]. Histopathological and animal model studies traditionally pointed to the spread of tau in a prion-like fashion through synaptic connections[14-19]. Others pointed to tau spread via brain hubs to functionally connected regions[20-23], showing similar accumulation rates between the co-activating regions[24-26]. Despite this mounting evidence that brain connectivity serves as a scaffold to tau accumulation in specific regions or networks, prior work linking tau to the different scales of brain connectivity has notable limitations. First, functional or structural connectomes have been mainly studied in isolation. Second, connectivity or pathology has traditionally been quantified via atlases with a limited number of discrete regions/parcels. Such parcellations inaccurately assume that a given (atlas-delineated) region is homogenous in its cytoarchitecture, neuronal connectivity, and/or function[27]. Third, current imaging frameworks typically apply simplified linear reductions to high-dimensional data or assume linear relationships between biomarkers. However, similar to music being composed of a superposition of sine waves, brain function exists as overlapping modes of connectivity variation[28,29] that can be non-linearly embedded in the brain's functional organization[30,31]. Finally, apart from these methodological limitations, it should be recognized that certain networks are almost invariably protected against tau despite their connectedness to high-tau regions (e.g., precuneus-sensorimotor circuit[32]). Moreover, tau starts and accumulates in a heterogeneous manner across different regions in various individuals. In this regard, it is imperative to consider that the different networks being progressively affected by AD pathology may be largely determined by underlying macro-scale axes of cortical organization in connectivity, microstructure, gene expression, and/or function.

To overcome the aforementioned limitations and shed light on whether AD pathological progression is related to the macro-scale organizational axes of the brain, we here contextualize the spatial maps of tau accumulation and neuroinflammation in AD to an emerging approach in understanding brain organization: gradients of connectivity[33]. A connectome gradient reflects the concept that patterns of connectivity within the brain vary systematically along a continuum. The most prominent connectivity gradient describes a smooth spatial transition in functional connectivity variation from unimodal (sensory-motor) to trans modal (default-mode) regions[34], which coincides with the brain's organizational axes from perception to abstraction[35]. In various clinical populations, gradient maps have shown alterations at the structural, functional or molecular level, and identified key subnetworks associated with disease, suggesting clinical utility for understanding and monitoring disease progression[36-41]. Despite the increasing value of the gradient approach in studying connectomics, no studies have yet investigated how both functional and structural gradients are altered in AD and relate to patterns of tau, neuroinflammation, and cognition.

In this work, we use subject-specific resting-state functional MRI (fMRI) and diffusion-weighted MRI (dMRI), along with molecular (PET) imaging, within a highly characterized dementia clinic cohort. We show that the organization of the brain, as uniquely described by gradients, plays a role in shaping the distribution of AD-related tau pathology. Specifically, we show that connectome gradients (i) are altered in AD, (ii) align with pathological (PET-derived) gradients and play a role in facilitating pathology progression, and (iii) interact with tau to affect cognition.

## Results

We studied 213 highly-characterized participants from the Translational Biomarkers in Aging and Dementia (TRIAD) cohort, of which 103 were Aβ-negative cognitively normal (CN A-, henceforth referred to as controls) and 110 were Aβ-positive diagnosed as either CN (CN A+; n = 35) or cognitively impaired with mild cognitive impairment or AD dementia (CI; n = 75). All participants underwent $^{18}$F-MK6240 tau-PET, $^{18}$F-NAV4694 Aβ-PET, structural MRI, dMRI, fMRI, APOE-ε4 genotyping, and a comprehensive cognitive battery. A subset of the cohort additionally underwent $^{11}$C-PBR28 neuroinflammation (TSPO) PET (n = 94, of which 50 were A+) and follow-up tau-PET (n = 87, of which 39 were A+; average follow-up from baseline scan: 420 ± 79 days). All demographic data are reported in Table 1. Sex and level of education were similar between groups. The CN A+ group was slightly older compared to the other groups (P = 0.02), while CI contained more APOE-ε4 carriers and had lower cognitive performance (P < 0.001). Both CN A+ and CI had elevated Aβ in the neocortex and tau in the Braak I–II regions, while CI also had elevated tau in Braak III–VI regions, compared to controls (P < 0.001). Details on inclusion and exclusion criteria, image processing, gradient extraction, cognitive composites, and statistical experiments are reported in the Methods.

### Connectome gradients reveal brain-wide network reorganization in AD

We investigated the functional and structural organization of the neocortex by decomposing the connectivity matrices into low-dimensional components (gradients) using diffusion embedding[34,42] (Fig. 1). In such representation, regions (nodes) with similar patterns of connectivity to the rest of the brain are located close together on the gradient while nodes with distinct patterns are located further apart. Gradients, thus, capture the dominating spatial patterns of connectivity variation that change gradually along a continuum.

The first and second gradients of functional connectivity (G1$_{FC}$ and G2$_{FC}$) explained 71% of the information (variance) in the first ten components. G1$_{FC}$ and G2$_{FC}$ distinguished sensory-motor from default-mode network (DMN) cortices (unimodal-transmodal gradient) and auditory from visual cortices (auditory-visual gradient), respectively, on their outer ends (Fig. 2a). These gradient patterns were visually similar across diagnostic groups (Supplementary Fig. 1a; for inter-subject variability maps see Supplementary Fig. 2a). Between-group comparisons of G1$_{FC}$ scores in CI individuals, compared to controls, revealed significantly reduced network segregation. Here, the unimodal and transmodal networks of G1$_{FC}$ moved closer to each other (i.e., becoming more similar in their brain-wide connectivity profile), leading to an overall gradient contraction in CI (Fig. 2b–d). This finding was robust across hemispheres (Supplementary Fig. 3, 4), gradient realignment strategies (cohort- vs. group-representative connectomes to which individual manifolds were aligned; Supplementary Fig. 5), and brain atlases (Supplementary Fig. 6). These between-group differences in G1$_{FC}$ were correlated with meta-analytic cognitive terms from perception (e.g., sensory, motor) to abstraction (e.g., theory of mind, semantic, retrieval) (Fig. 2d). Between-group comparisons of the higher-order functional gradients showed robust G2$_{FC}$ alterations in dorsal attention and sensory-motor networks (the former already in CN A+ participants compared to controls; Supplementary Fig. 5).

For structural connectivity, the first and second gradients (G1$_{SC}$ and G2$_{SC}$, explaining 48% of the information) distinguished temporo-

## Table 1 | Demographics

| | Overall<br>n = 213 | CN A-<br>n = 103 | CN A+<br>n = 35 | CI A+<br>n = 75 | P-Value[*] |
|---|---|---|---|---|---|
| **CN**, n (%) | 138 (64.8) | 103 (100.0) | 35 (100.0) | | |
| **MCI**, n (%) | 41 (19.2) | | | 41 (54.7) | |
| **AD dementia**, n (%) | 34 (16.0) | | | 34 (45.3) | |
| Sex female, n (%) | 129 (60.6) | 63 (61.2) | 24 (68.6) | 42 (56.0) | 0.447 |
| Age, mean (SD) | 69.4 (9.2) | 68.7 (9.7) | 73.5 (7.8) | 68.3 (8.8) | 0.012[a] |
| Education, mean (SD) | 15.3 (3.8) | 15.8 (4.1) | 14.6 (3.6) | 15.1 (3.5) | 0.219 |
| *APOE*-ε4 carriers, n (%) | 77 (36.8) (n = 209) | 27 (26.5) | 9 (25.7) | 41 (56.9) | |
| MMSE, mean (SD) | 27.4 (4.2) (n = 211) | 29.1 (1.2) | 28.9 (1.2) | 24.3 (5.8) | <0.001[b] |
| **Cognitive composites[♯]** | | | | | |
| Delayed memory, z (SD) | 0.7 (0.2) (n = 164) | 0.8 (0.1) | 0.7 (0.1) | 0.5 (0.2) | <0.001[b] |
| Immediate memory, z (SD) | 0.8 (0.2) (n = 172) | 0.8 (0.1) | 0.8 (0.1) | 0.6 (0.2) | <0.001[b] |
| Language, z (SD) | 0.8 (0.1) (n = 180) | 0.8 (0.1) | 0.8 (0.1) | 0.7 (0.2) | <0.001[b] |
| Speed, z (SD) | 0.6 (0.1) (n = 174) | 0.6 (0.1) | 0.6 (0.1) | 0.5 (0.1) | <0.001[b] |
| Executive function, z (SD) | 0.6 (0.2) (n = 176) | 0.7 (0.1) | 0.7 (0.1) | 0.5 (0.3) | <0.001[b] |
| Object recognition, z (SD) | 0.9 (0.1) (n = 175) | 0.9 (0.0) | 0.9 (0.0) | 0.9 (0.1) | <0.001[b] |
| Cognitive flexibility, z (SD) | 0.6 (0.2) (n = 169) | 0.6 (0.1) | 0.6 (0.1) | 0.4 (0.2) | <0.001[b] |
| Word reading, z (SD) | 0.7 (0.1) (n = 175) | 0.8 (0.0) | 0.8 (0.1) | 0.7 (0.1) | <0.001[b] |
| Cortical Aβ SUVR, mean (SD) | 1.7 (0.5) | 1.2 (0.1) | 1.8 (0.3) | 2.2 (0.4) | <0.001[c] |
| Tau SUVR (Br I-II), mean (SD) | 1.3 (0.6) | 0.9 (0.1) | 1.1 (0.3) | 1.8 (0.6) | <0.001[c] |
| Tau SUVR (Br III-IV), mean (SD) | 1.3 (0.7) | 0.9 (0.1) | 1.0 (0.2) | 1.8 (0.9) | <0.001[b] |
| Tau SUVR (Br V-VI), mean (SD) | 1.2 (0.5) | 1.0 (0.1) | 1.0 (0.2) | 1.6 (0.7) | <0.001[b] |

[*] *P*-values were based on ANOVA with Bonferroni correction and Tukey's post-hoc testing.
[a] CN A- or CI A+ vs. CN A+: P = 0.02; [b] CN A- or CN A+ vs. CI A+: P ≤ 0.001; [c] all groups: P ≤ 0.002.
[♯] Individual raw scores were standardized following the method described in Malek-Ahmadi et al. 2018 and averaged across subdomains to create the composite scores. See "Methods" for details.

occipital from frontal cortices (posterior-anterior gradient) and sensory-motor from limbic cortices (superior-inferior gradient), respectively, on their outer ends (Fig. 2e). The overall gradient patterns were similar across diagnostic groups (Supplementary Fig. 1b; for inter-subject variability maps see Supplementary Fig. 2b). Group-wise differences did not indicate a global gradient contraction as for functional connectivity, and they were less consistent across hemispheres (Fig. 2f–h; Supplementary Fig. 7, 8), gradient realignment strategies (Supplementary Fig. 9), or atlases (Supplementary Fig. 10). Notably, robust group-wise differences involved $G1_{SC}$ and $G2_{SC}$ reorganization of early tau-accumulating regions (cingulate, temporal and orbitofrontal regions; Supplementary Fig. 9). $G1_{SC}$ differences were correlated with meta-analytic cognitive terms from perception (e.g., pain, touch) to memory (e.g., episodic memory, face recognition) (Fig. 2h).

For both functional and structural connectomes, while G1 and G2 explained most of the information, the explained information dropped to <10% for higher-order gradients (Supplementary Fig. 1c).

Sensitivity analyses further validated the robustness of our findings. First, we regrouped participants based on their biomarker profile only (amyloid [A] and tau [T] PET positivity), irrespective of cognitive status. Functional and structural gradients were most prominently altered in A+ T+, and similar networks were affected as in our main analysis (Supplementary Fig. 11a). Second, we replicated gradient patterns using different connectome thresholding and similarity kernels (Supplementary Fig. 11b). Notably, an alternative kernel based on normalized angle explained lower information in the structural connectome data ($G1_{SC}$: 14%, $G2_{SC}$: 13%) compared to the employed cosine similarity ($G1_{SC}$: 28%, $G2_{SC}$: 20%). Third, the observed regions of sex differences (Supplementary Fig. 11c) were largely located within the medial parieto-temporal cortex of the DMN for $G1_{FC}$, with females showing lower scores than males (i.e., moving more transmodal). While, for $G1_{SC}$, females showed lower scores mainly in cingulo-opercular and somatomotor cortices.

### Connectome gradients relate to pathology distribution

To test the hypothesis that connectivity gradients play a role in shaping the distribution of AD-related pathology, we extracted PET gradients for $^{18}$F-MK6240 (tau-PET) and $^{11}$C-PBR28 (inflammation-PET) from group-level SUVR covariance matrices and assessed their relation to connectivity gradients (Fig. 1). We found that both $G1_{FC}$ and $G1_{SC}$ significantly correlated with the organization of tau-PET along its gradients ($G_{TAU}$) (Fig. 3a). Specifically, $G1_{TAU}$ (explaining 40% of the information in CI) was closely aligned with $G1_{FC}$ (Spearman ρ = 0.68, $P_{adj}$ < 0.001; Fig. 3b), indicating that regions with a more similar brain-wide functional connectivity profile had a more similar brain-wide tau-PET covariance profile. In contrast to $G1_{TAU}$, $G2_{TAU}$ (28% of the information in CI) highly overlapped with the Braak stages (Fig. 3a-right) and was aligned with $G1_{SC}$ (ρ = 0.70, $P_{adj}$ < 0.001; Fig. 3c) but not $G1_{FC}$. Notably, this correlation of $G2_{TAU}$ with $G1_{SC}$ was already prominent in the CN A+ participants (Supplementary Fig. 12). Taken together, this suggests a role for structural connectivity in shaping early-stage tau distribution, while functional connectivity may play a more prominent role in shaping tau distribution in the CI stage (at the cross-sectional level).

With regards to neuroinflammation, the alignment of $G1_{INFLAM}$ with $G1_{FC}$ was significant but less prominent than with $G1_{SC}$ (ρ = 0.39 [$P_{adj}$ = 0.008] vs. ρ = 0.72 [$P_{adj}$ < 0.001] in CI; Fig. 3e, f). $G1_{INFLAM}$ also closely overlapped with the Braak stages (Fig. 3d-right). This indicates that regions with a more similar brain-wide structural connectivity profile had a more similar brain-wide inflammation-PET covariance profile, both in CI and CN A+ (at the cross-sectional level) (Supplementary Fig. 13).

While the G1 and G2 explained most of the information in PET data (tau: 68%, inflammation: 58%), the explained information dropped substantially for higher-order gradients (Supplementary Fig. 14). Connectome-PET correlations were weaker among higher-order gradients (Supplementary Fig. 15a, b). Sensitivity analyses using different

**a Input data**
*MULTI-MODALITY IMAGE PROCESSING*

**b Gradient identification**
*PER MODALITY*

**c Analyses**

Fig. 1 | **Methodology of gradient generation. a** Our multi-modal input data included dMRI, fMRI, and PET images, co-registered to a high-resolution custom brain atlas. **b** The resulting modality-specific connectomes or covariance matrices were transformed into a similarity matrix and subjected to diffusion map embedding. The resulting gradients make up a low-dimensional coordinate space. The interpretation of gradients (as applied in the current work) is visually compared to the traditional atlasing techniques, showing overlapping modes of connectivity similarity vs. discrete regions. **c** The main analyses involve either cohort-level investigations using group-wise connectomes/covariance matrices (Spearman's rank coefficient ρ between different modalities) or individual participants' connectomes (between-group differences based on t-statistics).

covariance thresholding and similarity kernels are reported in Supplementary Fig. 16. Importantly, our gradient method via diffusion embedding outperformed linear dimensionality reduction via principal component analysis (PCA) in terms of explained variance of the extracted components or latent variables, and the association between the resulting connectome and PET components (Supplementary Fig. 17).

**Tau deposition within connectivity hubs drives tau accumulation**

To further decode the effects of connectivity on tau accumulation, we performed a similar gradient-based analysis employing longitudinal tau-PET gradients. $G1_{\Delta TAU}$ (explaining 51% of the information in A+) distinguished between temporal and sensory-motor cortices (temporal-unimodal gradient) on its outer ends (Fig. 4a). $G1_{\Delta TAU}$ overlapped with the Braak stages (Fig. 4b-right) and was aligned with $G1_{SC}$ (ρ = 0.46, $P_{adj} < 0.001$; Fig. 4c) and $G1_{INFLAM}$ (ρ = 0.60, $P_{adj} < 0.001$; Fig. 4d, e). Conversely, $G2_{\Delta TAU}$ (13%) was more closely aligned with $G1_{FC}$ (ρ = 0.36, $P_{adj} < 0.001$) than with $G1_{SC}$, and did not overlap with the Braak stages. These correlations were robust across hemispheres (connectome-ΔPET correlations among the first ten gradients are displayed in Supplementary Fig. 15c).

Previous work has alluded to the role of tau epicenters, as well as the effects of either functional or structural connectivity in driving tau progression[24,43,44]. However, to our knowledge, no studies have yet incorporated both subject-specific functional and structural connectomes together to model the effects of connectivity and network hubs on in-vivo longitudinal tau accumulation. We leveraged the connectome gradient space as a low-dimensional coordinate space that is sensitive to topological changes of subject-specific connectomes. Our

initial step involved identifying, in gradient space, the nearest connected (based on shortest Euclidean distance) brain hubs (based on 'degree') for each ROI at the subject level (Fig. 5a). Once each ROI's gradient-derived hubs were identified, we next computed the average tau SUVR within these hubs (Fig. 5a). We then investigated whether tau within these hubs was associated with longitudinal tau accumulation (for all ROIs), accounting for baseline tau within the ROI and age, sex, and *APOE-ε4* status. Our findings, depicted in Fig. 5b, showed that longitudinal tau accumulation in ROIs of early tau buildup (MTL and lateral temporo-occipital cortices) was facilitated by high tau deposition within their structural, but not functional, hubs. Conversely, tau accumulation in ROIs of the frontoparietal cortex was facilitated by high tau deposition within their functional, but not structural, hubs. To further decode this finding, we then selected the top two positive and negative t-statistic ROIs and averaged the tau within their gradient-derived hubs (Fig. 5c); where, the temporo-occipital or frontoparietal ROIs showed lower tau SUVR than their higher-tau structural or functional hubs, respectively. This may be interpreted as high-tau hubs driving future tau accumulation in their lower-tau connections (in gradient space). We replicated our findings within the A+ group but not the A- group (Supplementary Fig. 18). Finally, we performed leave-one-out cross-validation to assess how well our model predictions matched the observed data (mean relative RMSE ~ 15%, Supplementary Fig. 19). We also repeated our main analysis using tau within FC or SC hubs as single predictors of tau accumulation (instead of together in the model as our main analysis), showing similar findings (Supplementary Fig. 20a). Adding baseline Aβ SUVR as a covariate also did not significantly alter the results (Supplementary Fig. 21).

We further validated our findings by testing the relative importance of hub selection and gradient-based connectivity in the model.

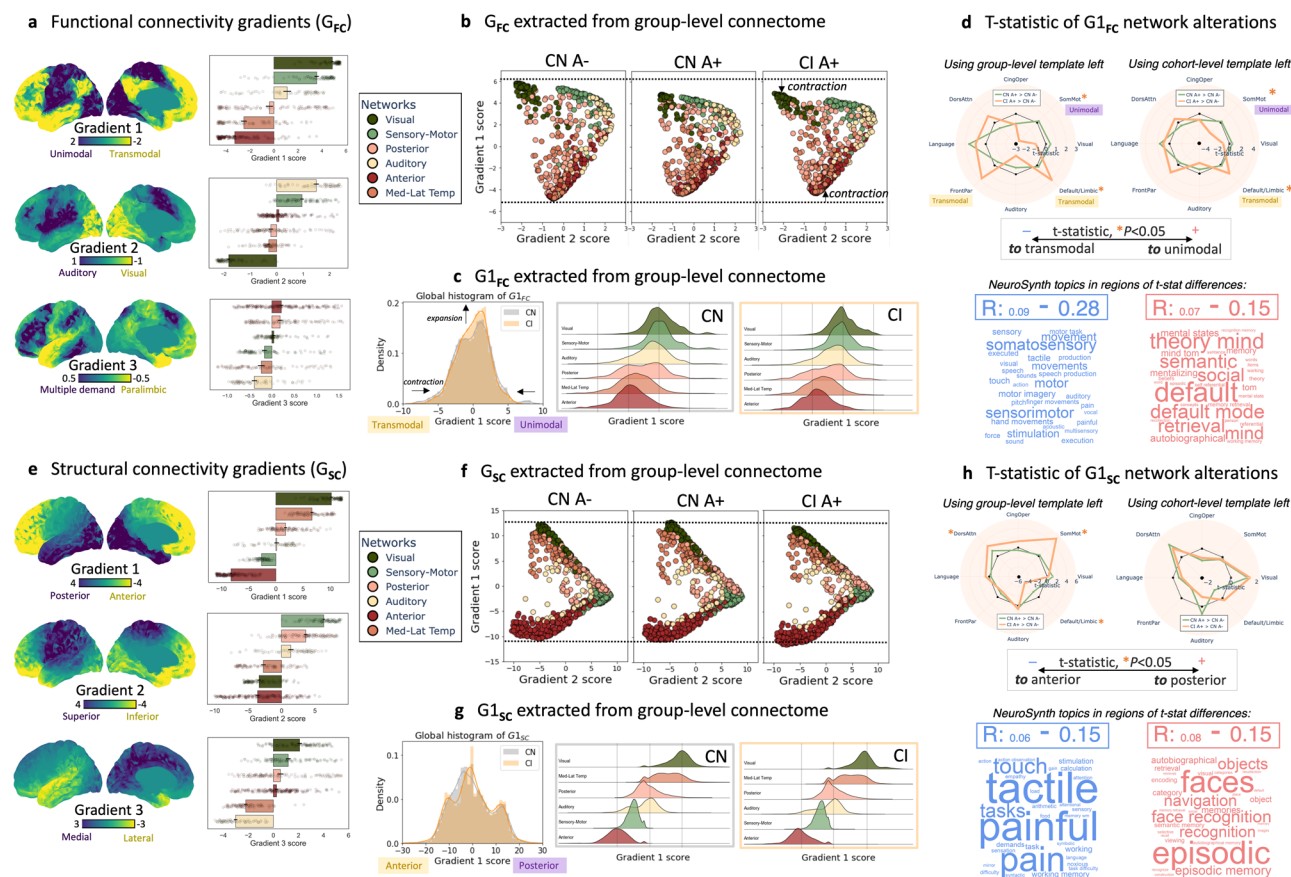

**Fig. 2 | Functional and structural connectivity gradients are altered in AD. a** The first three functional connectivity gradients (G_FC) projected onto the left-brain surface, extracted from the cohort-level connectome (average of n = 213 connectomes). Similar colors along the purple-yellow scale represent similar brain-wide connectivity patterns. The bar plots represent the corresponding network-specific average gradient scores ± standard error of the mean (SEM). G1_FC, G2_FC, and G3_FC explained 55, 17, and 9% of the information in functional connectome data, respectively. **b** Coordinate system spanned by the first two gradients based on group-level functional connectomes (average of n = 103, 35, and 75 connectomes, respectively), indicating G1_FC contraction with unimodal and transmodal (DMN) regions moving closer to each other in CI participants. **c** Histogram of gradient scores reflected global G1_FC contraction with an expansion of scores centered around zero. **d** Between-group comparisons (green: CN A+ [n = 35] vs. controls [n = 101]; orange: CI [n = 72] vs. controls [n = 101]) of network-based G1_FC alterations (asterisks represent significant t-statistics at the network-level with two-sided P < 0.05, adjusted for age, sex, and *APOE-ε4*) using a group-level (left) or full cohort-level (right) gradient realignment strategy. Word clouds of NeuroSynth cognitive terms associated with regions with positive (red) or negative (blue) t-statistic G1_FC differences between diagnostic groups (using cohort-level realignment strategy). **e** G1_SC, G2_SC, and G3_SC explained 28, 20, and 9% of the information in structural connectome data, respectively. **f** Coordinate system spanned by the first two gradients. **g** Histogram of G1_SC. **h** Between-group comparisons (green: CN A+ [n = 35] vs. controls [n = 102]; orange: CI [n = 72] vs. controls [n = 102]) of network-based G1_SC alterations and corresponding word clouds. Results are displayed for the left hemisphere; Supplementary Fig. 1, 3, and 7 show right hemisphere projections and group differences. Source data are provided as a Source Data file. Abbreviations: CN cognitively normal, CI cognitively impaired, FC functional connectivity, G gradient, SC structural connectivity.

First, we computed the average burden of tau within each ROI's nearest connections, irrespective of that connection being a degree hub. We observed that tau accumulation within an ROI was associated with high tau deposition within the ROI's nearest connections, irrespective of hub status (Supplementary Fig. 20b). Thus, distance in gradient space is a strong predictor of tau accumulation. In contrast, when selecting non-hubs with weak gradient-based connectivity to the ROI, effect sizes were weaker (Supplementary Fig. 20c). Notably, when we repeated our analysis using raw FC and SC gradient distances as predictors of tau accumulation (rather than tau within these connections), only increased baseline tau level (but not FC or SC) was associated with faster tau progression (Supplementary Fig. 20d). Supplementary Fig. 22a depicts unthresholded t-stats maps of the main analysis.

### Connectome gradients interact with tau to drive cognitive impairment

We finally investigated whether regional tau and connectome gradient scores had interacting effects on both baseline and 2-year changes in cognition. Both at baseline (Fig. 6a) and longitudinally (Fig. 6c), MMSE and language scores were negatively associated with G1_FC score among transmodal regions with greater tau SUVR, while positively associated with G1_FC score among unimodal regions with greater tau SUVR. Results are shown in A+ participants, with similar findings across all participants. Figure 6b–d shows the results for G1_SC and Supplementary Fig. 22b shows unthresholded t-stats maps. We performed leave-one-out cross-validation to assess how well our model predictions matched the observed data (mean relative RMSE ~ 16%, Supplementary Fig. 23). Exploratory results for other cognitive domains are shown in Supplementary Fig. 24 (strongest effects for object recognition). As with G1, the higher-order gradients likewise showed a significant interaction between tau and gradient score on cognitive impairment (Supplementary Fig. 25), though effects were weaker compared to G1. No significant interaction effects between connectome gradient scores and inflammation-PET on cognition were observed.

To further dissect these tau-connectome interactions on cognition, we investigated whether the effects of tau on cognition changed in a topology-dependent manner along the cortical hierarchy. We contrasted results within gradient-derived meta-ROIs (representing

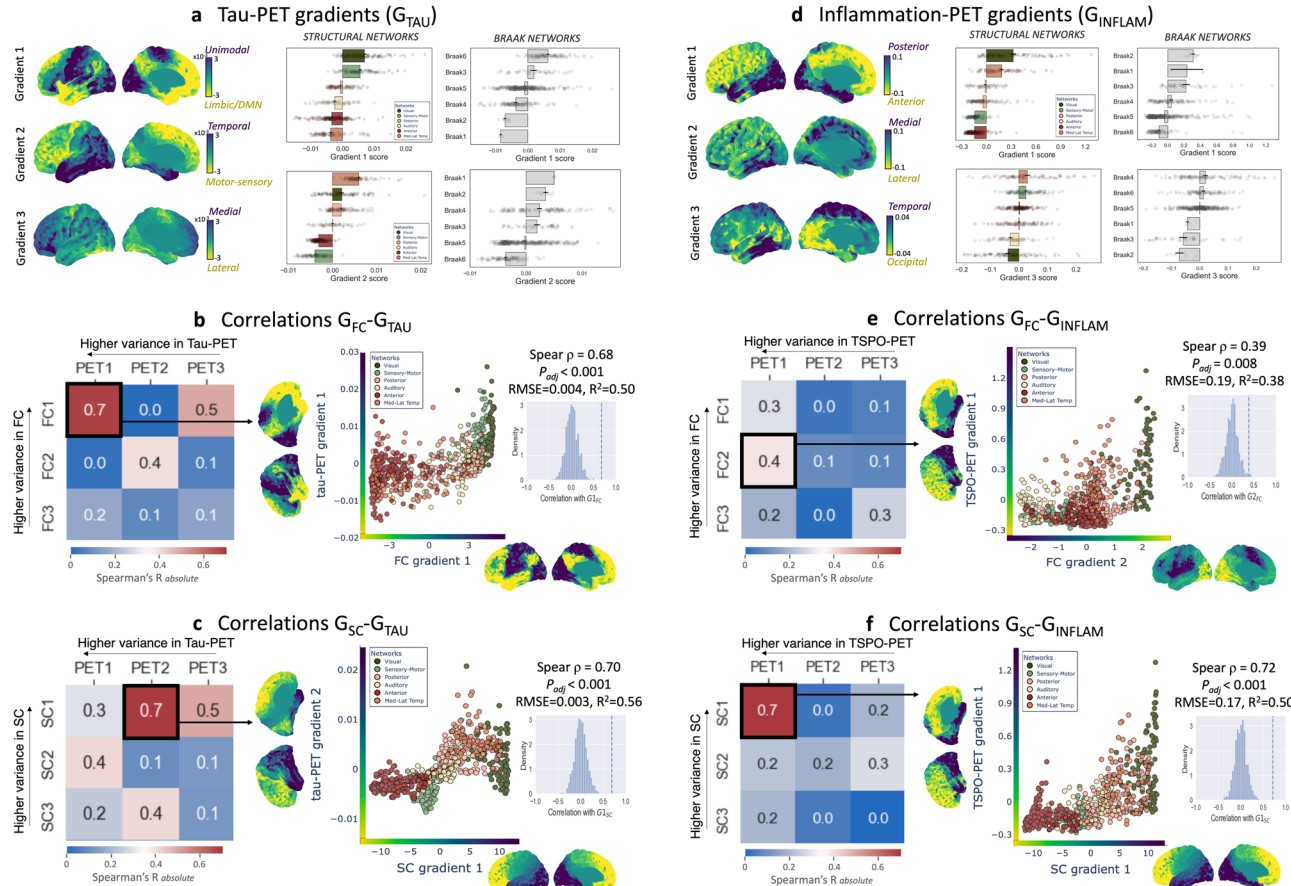

**Fig. 3 | Connectivity gradients align with PET gradients. a** The first three gradients of tau-PET ($G_{TAU}$) projected onto the brain surface and their corresponding network-specific average values ± SEM for $G1_{TAU}$ and $G2_{TAU}$ (40 and 28% explained the information in tau-PET data), extracted from the CI group (average of n = 75 tau covariance matrices). **b** Heatmap showing Spearman's rank correlations ρ between gradients of functional connectivity ($G_{FC}$) and $G_{TAU}$, indicating a strong association between their primary gradients $G1_{FC}$ and $G1_{TAU}$. **c** ρ between gradients of structural connectivity ($G_{SC}$) and $G_{TAU}$, indicating a strong association between $G1_{SC}$ and $G2_{TAU}$. **d** The first three gradients of inflammation (TSPO)-PET ($G_{INFLAM}$) projected onto the brain surface and their corresponding network-specific average values for $G1_{INFLAM}$ and $G3_{INFLAM}$ (39 and 11% explained information in TSPO-PET data), extracted from the CI group (thresholded at 50% sparsity due to low sample size

[average of n = 32 TSPO covariance matrices]). $G2_{INFLAM}$ (19% explained information) may primarily reflect partial volume effect and was not included in the bar-plots. **e** Heatmap showing ρ between $G_{FC}$ and $G_{INFLAM}$, indicating a modest association between $G2_{FC}$ and $G1_{INFLAM}$. **f** ρ between $G_{SC}$ and $G_{INFLAM}$, indicating a strong association between $G1_{SC}$ and $G1_{INFLAM}$. Results are displayed for the left hemisphere; Supplementary Fig. 14, 15 show right hemisphere projections and correlations. A cubic polynomial was fitted for each of the regressions, indicating absolute RMSE and $R^2$ of the fitted model. The two-sided P-value of gradient correlations was tested using null models using spatial autocorrelation-preserving surrogates based on variogram matching (1000 permutations). Source data are provided as a Source Data file. Abbreviations: CI cognitively impaired, FC functional connectivity, G gradient, RMSE root-mean-square error, SC structural connectivity.

gradual transitions between composite subnetworks) to those based on the predefined Braak regions. First, we found that tau was significantly associated with cognition across the entire principal gradient (Fig. 7a). Conversely, only a few Braak-based tau-cognition associations reached R > 0.6 (none for Braak I-II). Second, the strength of these associations changed in a topology-dependent manner progressively along the gradient axes (Fig. 7b). The associations were generally strongest in the higher-order transmodal/anterior regions while weakest in unimodal/posterior regions. For example, tau correlated with non-memory domains gradually along $G1_{FC}$ but not the Braak axis (e.g., cognitive flexibility in A+: $R^2 = 0.58$ vs. Braak $R^2 = 0.38$). Similarly, tau correlated with memory gradually along $G1_{SC}$ but not the Braak axis (e.g., delayed memory in A+: $R^2 = 0.95$ vs. Braak: $R^2 = 0.28$). Sensitivity analyses with different numbers of gradient bins (meta-ROI sizes) yielded similar results (Supplementary Fig. 26).

Last, we performed a NeuroSynth meta-analytic decoding of each of our primary template gradients (Fig. 7c and Supplementary Fig. 27), showing how cortical organization underlies cognitive functions. We observed that cognitive domains are associated with the gradient

organization from the unimodal (e.g., motor, sensory perception) to the transmodal (e.g., social, negative emotion, moral, memory) poles of both the functional and tau-PET primary gradients (Fig. 7c). While, cognitive terms of facial recognition, sensory perception, and memory are also organized at the posterior poles of both the structural, inflammation and Δtau-PET gradients (Supplementary Fig. 27). Taken together, our results suggest that connectivity (gradient)-derived meta-ROIs represent a sensitive method to capture brain-behavior relationships.

## Discussion

Certain brain subnetworks exhibit greater similarity in their extent and rate of pathology accumulation. Such similarity may be linked to those regions' unique apical position in the cortical hierarchies, which can be interrogated in a data-driven manner at the system level through mapping of connectivity gradients[33,34,45]. In this work, we decomposed high-dimensional data of both functional and structural connectivity into its main axes of variance (gradients) at the subject level and situated patterns of tau or inflammation along these organizing axes.

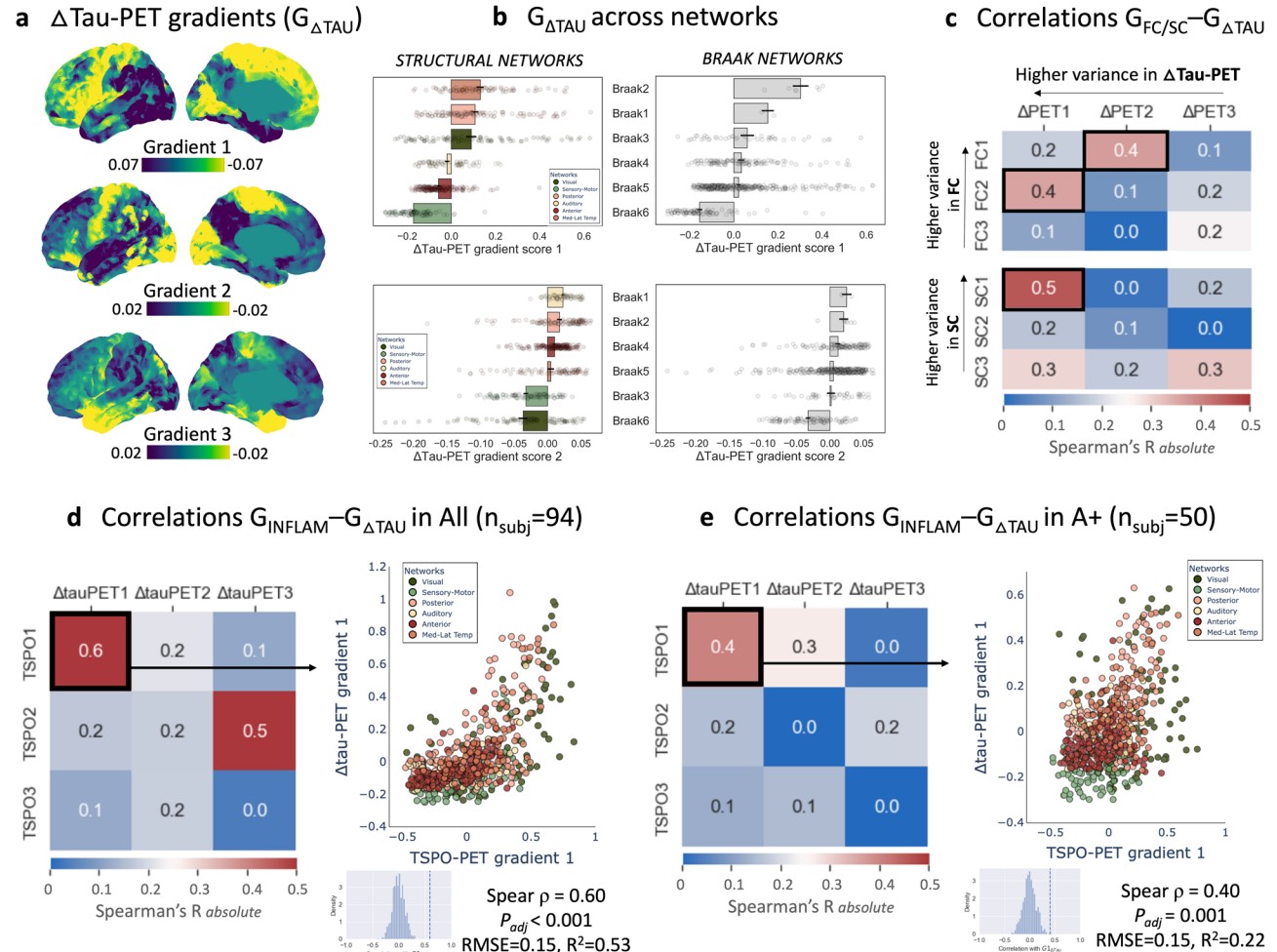

**Fig. 4 | Longitudinal tau-PET gradients align with connectome and inflammation gradients. a** The first three gradients of longitudinal tau accumulation ($G_{\Delta TAU}$) projected onto the left brain surface, extracted from the A+ group (average of n = 39 Δtau covariance matrices). **b** Network-specific average values ± SEM for $G1_{\Delta TAU}$ (top) and $G2_{\Delta TAU}$ (bottom), explaining 51 and 13% of the information respectively in Δtau-PET data, with the first gradient largely overlapping with Braak stages. **c** Heatmap showing Spearman's rank correlation ρ between $G_{FC}$ or $G_{SC}$ and $G_{\Delta TAU}$, indicating good alignment ($G1_{FC}$-$G2_{\Delta TAU}$: RMSE = 0.04, $R^2$ = 0.26; $G2_{FC}$-$G1_{\Delta TAU}$: RMSE = 0.16, $R^2$ = 0.12; $G1_{SC}$-$G1_{\Delta TAU}$: RMSE = 0.15, $R^2$ = 0.30). Results are

displayed for the left hemisphere; Supplementary Fig. 15 shows right hemisphere correlations. **d** ρ between $G_{\Delta TAU}$ and $G_{INFLAM}$ based on template gradients across the cohort. **e** ρ between $G_{\Delta TAU}$ and $G_{INFLAM}$ based on template gradients in A+. The two-sided P-value of gradient correlations was tested with null models using spatial autocorrelation-preserving surrogates based on variogram matching (1000 permutations). Source data are provided as a Source Data file. Abbreviations: FC functional connectivity, G gradient, RMSE root-mean-square error, SC structural connectivity.

---

Our findings are: 1) AD disrupts the topological organization of the brain by focally altering the connectome gradients, with a notable contraction of the unimodal-transmodal functional axis; 2) AD-related gradient contraction interacts with tau to predict cognitive decline; 3) group-level gradients unveil spatial similarity between brain-wide patterns of connectivity and AD pathology, with the primary structural gradient (from tractography) showing good alignment with future tau accumulation patterns; and 4) subject-level gradients can be employed to explain future tau accumulation, such that temporo-occipital tau accumulation is facilitated by high tau within gradient-derived structural hubs while frontoparietal tau accumulation is facilitated by high tau within gradient-derived functional hubs. Taken together, these results suggest that the principal axes of functional and structural organization of the neocortex as mapped by gradients play a role in shaping the distribution and progression of AD-related pathology. This work demonstrates that the principal gradients represent an intrinsic coordinate system to predict tau accumulation in single individuals.

The gradient approach has proven utility in several models of health and disease including neurodevelopment[46–48], aging[38], and neurological/neuropsychiatric conditions such as stroke[39], autism[37],

epilepsy[49], and major depressive disorder[50,51]. Functional gradients have served as sensitive biomarkers of treatment response in several neuropsychiatric disorders[51,52]. Our study adds to this growing field of gradient-based applications by investigating connectivity gradients (both functional and structural) and their relation to pathology in AD. The gradient technique reduces the number of data points from NxN nodes (representing region-to-region connectivity) into a series of Nx1 spatial maps (representing region-to-region similarity in brain-wide connectivity patterns). While a typical (NxN) connectivity matrix is restricted to the connections between specific pairs of regions, our brain-wide similarity measure explains the region's global role in multiple networks. The derived principal gradients follow established models of cortical hierarchy and laminar differentiation[35].

In AD, we observed an overall unimodal-transmodal functional gradient contraction, an axis most consistent with cortical maturation and gradual change in several features of the neocortex[44] including gene expression (sensory-to-association[53]), myelination (high-to-low[54]), PET-derived neurotransmitter receptor profiles[55], glucose metabolism (low-to-high[40,41]), cerebral blood flow (low-to-high), and cognition (perception-to-abstraction[35]). This unimodal-transmodal

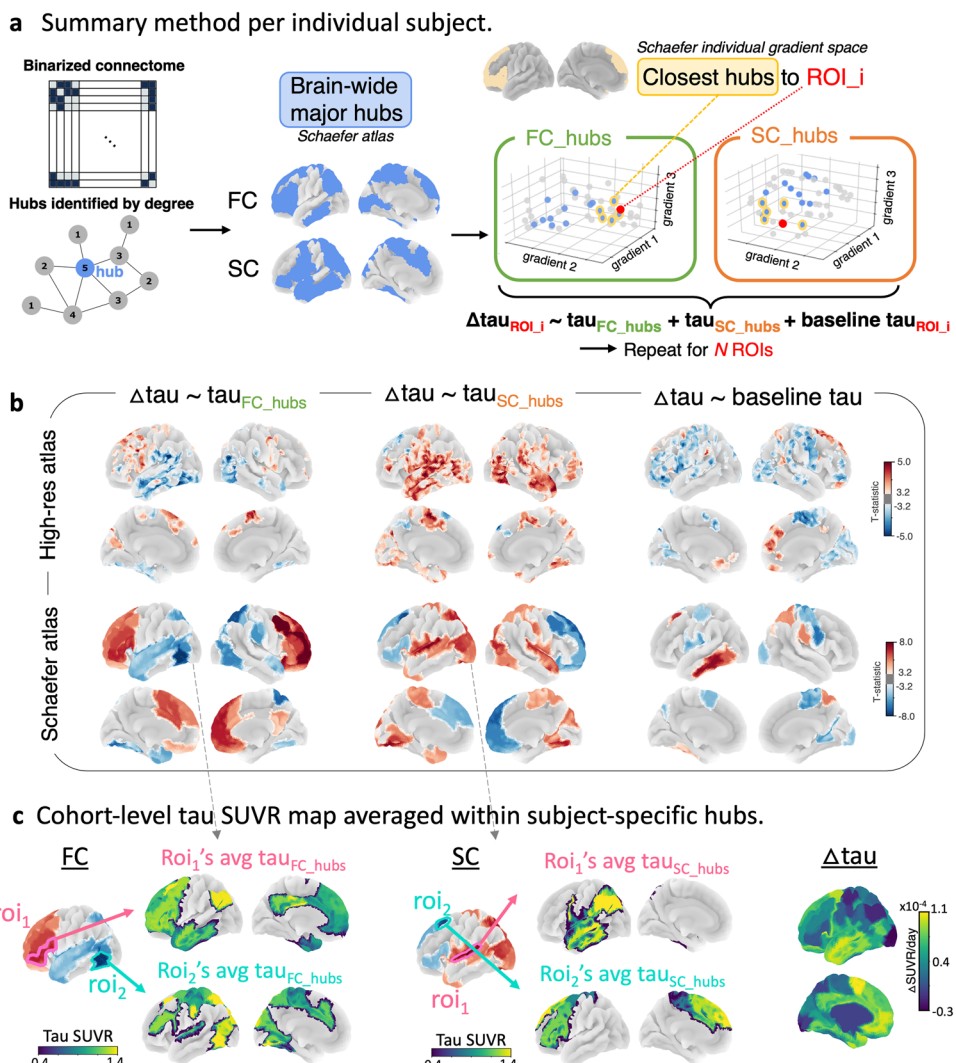

**Fig. 5 | Tau within gradient-derived subject-specific hubs drives tau accumulation. a** Subject-specific hubs were identified (blue) based on the graph theoretical metric called degree extracted from the thresholded and binarized connectivity matrix, from which the top nearest distance hubs (in gradient space) to each ROI were selected (yellow), for each participant. Tau SUVR was then averaged within each of these FC hubs (green; $\text{tau}_{\text{FC\_hubs}}$) and SC hubs (orange; $\text{tau}_{\text{SC\_hubs}}$). **b** Regression results (n = 86 participants) of longitudinal $\Delta\text{tau}_{\text{ROI}}$ with baseline $\text{tau}_{\text{FC\_hubs}}$ and $\text{tau}_{\text{SC\_hubs}}$, adjusted for age, sex, *APOE*-ε4, baseline $\text{tau}_{\text{ROI}}$, and FWE at two-sided P < 0.001, replicated for two different atlases. A positive t-statistic (red) within an ROI indicates a positive relationship between tau accumulation within the ROI and baseline tau within the ROI's hubs, while a negative t-statistic (blue) indicates a negative relationship. **c** Schematic showing the baseline tau within the subject-wise selected hubs nearest connected to the top positive and negative t-statistic ROI from panel **b**, averaged across the cohort. A positive t-statistic (red color in panel **b**) resulted from a relatively higher tau deposition within the ROI's hubs (see average [avg] $\text{tau}_{\text{FC\_hubs}}$ and $\text{tau}_{\text{SC\_hubs}}$) compared to tau within the ROI itself, while a negative t-statistic (blue color in panel **b**) resulted from a relatively lower tau deposition within the ROI's hubs. The average $\Delta\text{tau}$ SUVR among all participants is shown in the outer right panel. Source data are provided as a Source Data file. Abbreviations: FC functional connectivity, ROI region-of-interest, SC structural connectivity, SUVR standardized uptake value ratio.

contraction may reflect a loss of functional network segregation, possibly indicative of blurring of individual network specialization[56–59] (de-differentiation[60]). We also showed gradient changes along the visual-auditory functional axis (explaining less variance than the unimodal-transmodal axis), with early-stage reorganization of the dorsal attention network in correspondence with recent work[61]. On the other hand, our two primary structural gradients captured previously identified axes of cortical thickness[62,63] and neuron density[33] and showed only modest alterations in AD, mainly reorganization of early tau-accumulating regions (lateral/medial-temporal, cingulate, and orbitofrontal cortices).

In regard to connectivity-related tau-PET patterns, we observed that regions exhibiting greater similarity in their connections to the rest of the brain also exhibit greater similarity in how they co-vary in their tau distributions. In other words, spatial connectivity similarity

shapes spatial pathology (progression) similarity in AD. Previous in-vivo imaging studies typically promoted connectivity to tau epicenters as the main driver of tau spread. Notably, Franzmeier et al. demonstrated that regions with similar tau-PET accumulating rates were strongly functionally connected to each other using fMRI[20,24]. Schoonhoven et al.[64] instead used a single-epicenter epidemic spreading model based on MEG data to capture neuronal activity directly and demonstrated a superior role for functional over structural connectivity on tau spread. However, that model did not include the entorhinal cortex, a region identified as a common tau epicenter[65] with strong structural connections to the rest of the temporal lobe. Vogel et al.[65] found that the structural connectome (from a normative cohort) outperformed the functional connectome in simulating tau spread, with the strongest predictions in the earlier disease stages. Other in-vivo imaging studies with cross-sectional designs similarly

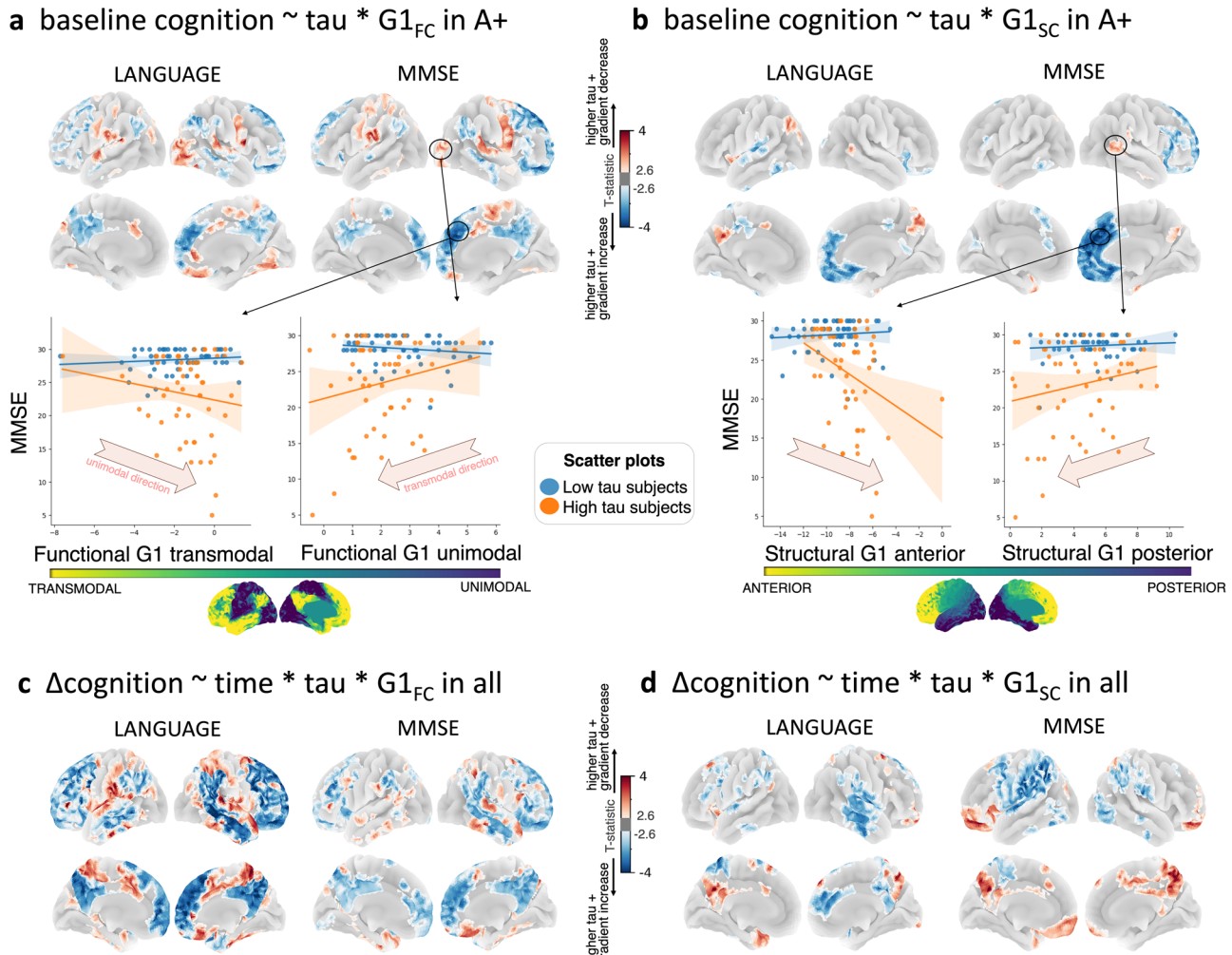

**Fig. 6 | Interaction between connectome gradient and tau on cognition. a** The interaction effect between regional tau SUVR and $G1_{FC}$ score (aligned to the full cohort-level template) on baseline MMSE and language in A+ (n = 107 and n = 90, respectively). A negative interaction effect (blue t-statistic in transmodal regions) indicates that higher tau together with a higher $G1_{FC}$ score (gradient contraction towards unimodal) results in lower cognition. Similarly, a positive interaction (red t-statistic in unimodal regions) indicates that higher tau together with lower $G1_{FC}$ score (gradient contraction towards transmodal) is associated with lower cognition. **b** The interaction effect between regional tau SUVR and $G1_{SC}$ score on cognition. The scatterplot illustrates the interaction effect between regional tau (binned into low [blue] vs. high [orange] for visualization) and G1 score for representative ROIs (see Source Data file), with linear fits and 95% confidence

intervals. **c, d** The 3-way interaction effect between time, regional tau SUVR and $G1_{FC}$ (panel **c**) or $G1_{SC}$ (panel **d**) on 2-year cognitive change across all participants. Sample sizes varied across composite scores: MMSE: $n_{visit1} = 104$, $n_{visit2} = 89$, $n_{visit3} = 58$ and language: $n_{visit1} = 99$, $n_{visit2} = 78$, $n_{visit3} = 58$. Limbic regions are blue for $G1_{FC}$ and red for $G1_{SC}$ because they are largely located on the negative vs. the more positive pole of the respective gradients; while, the prefrontal cortex (blue) is located on both the negative poles of the respective gradients. All analyses were adjusted for age, sex, education, *APOE*-ε4 and FWE at two-sided P < 0.01. Source data are provided as a Source Data file. Abbreviations: FC functional connectivity, G gradient, MMSE Mini-mental state examination, ROI region-of-interest, SC structural connectivity, SUVR standardized uptake value ratio.

highlighted close tau-axonal associations in a temporal-occipital-anterior pattern[66,67].

The current work, leveraging gradient-based (brain-wide) connectomics, suggests that tau may not be simply distributed along either functional or structural connections but instead follows connectivity gradients in a stage- and spatial-dependent manner, creating overlapping tau distributions. Specifically, in symptomatic patients with amyloid pathology, the primary gradient of baseline tau-PET variation was closely aligned with the unimodal-transmodal functional gradient. The secondary gradient of baseline tau-PET variation, as well as the primary gradient of Δtau-PET variation, aligned with the posterior-anterior structural gradient already from preclinical stages and more closely replicated the pathologic-defined Braak stages of longitudinal tau progression. Importantly, this primary structural gradient (from tractography) showed a better alignment with tau accumulation patterns than the primary functional gradient. Taken

together, this work suggests a role for the structural connectome gradient in shaping early tau distribution as well as its progression over time following a posteromedial-dominant pattern. In support of this notion, the MTL displays its strongest structural connections to the posterior but not prefrontal cortices[68] and forms a pathway of tau propagation to posteromedial structures of the DMN along the hippocampal cingulum bundle[66].

Our subject-tailored analysis lends further evidence to the notion of spatial dependence, revealing distinct patterns of tau accumulation within temporo-occipital vs. frontoparietal cortices. Specifically, higher rates of temporo-occipital tau accumulation were associated with higher levels of tau deposition within gradient-derived structural hubs, whereas frontoparietal tau accumulation was associated with higher tau within gradient-derived functional hubs. These hubs were extracted from individual connectome gradients. As tauopathy is thought to typically start in anatomically defined early-Braak areas, the

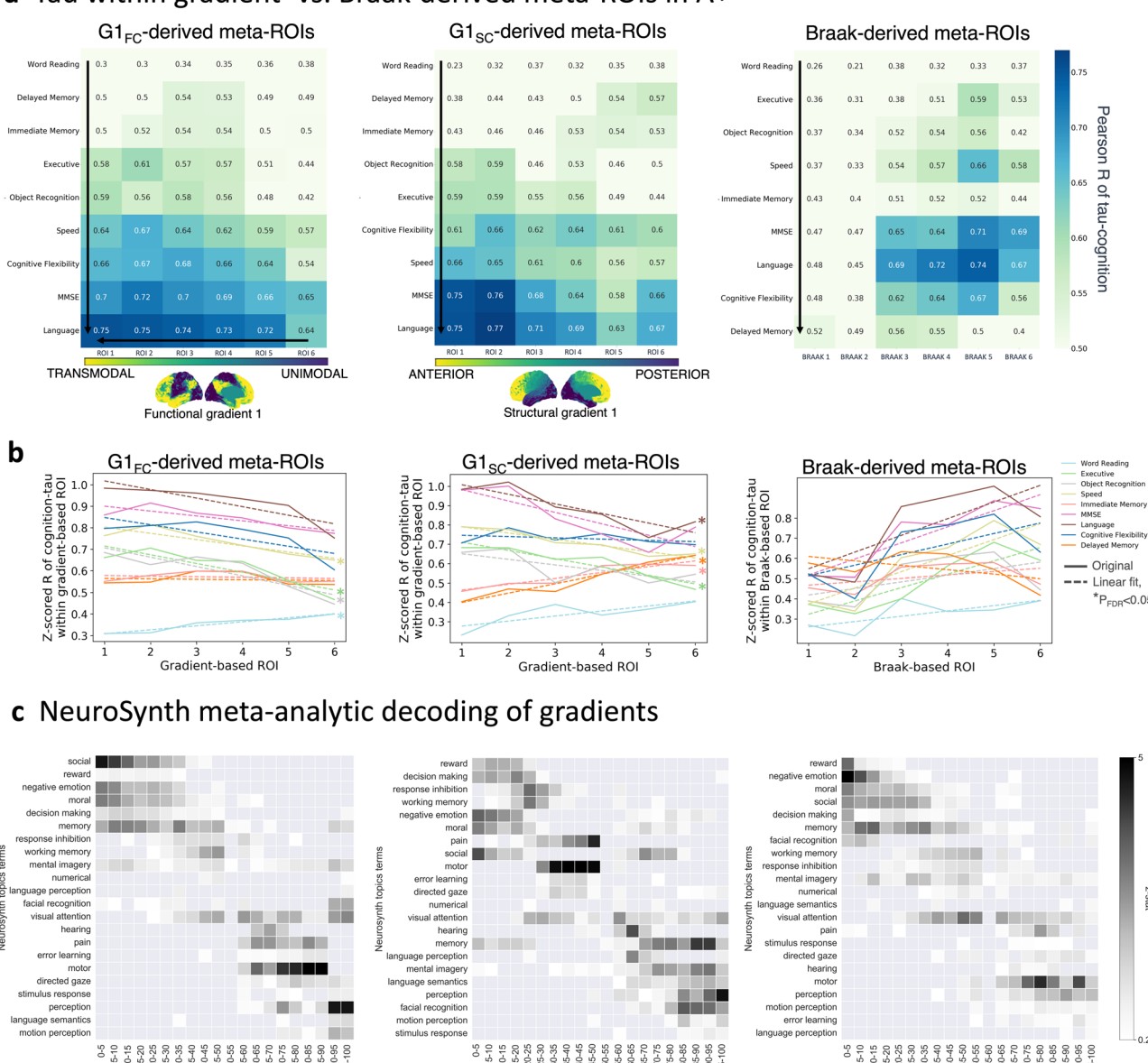

**Fig. 7 | Gradient-derived ROIs capture brain-behavior relationships. a** Cognitive correlates of tau SUVR within G1$_{FC}$-derived (left), G1$_{SC}$-derived (middle), or Braak meta-ROIs (right). Partial regression (absolute Pearson's R) adjusted for age, sex, education, and *APOE-ε4*. Sample sizes of A+ participants varied across composite scores: word reading $n = 87$, delayed memory $n = 76$, immediate memory $n = 82$, executive function $n = 86$, object recognition $n = 86$, processing speed $n = 84$, and cognitive flexibility $n = 81$. **b** The resultant correlation coefficients changed in a topology-specific manner along the G1$_{FC}$ and G1$_{SC}$ but not the Braak axes (based on a linear regression between gradient bin ordering and the tau-cognition [z-scored] correlation coefficient within each bin, at two-sided P$_{FDR}$ < 0.05). **c** Z-statistic maps of the associations between meta-analytic cognitive terms and our primary functional (left), structural (middle), and tau-PET (right) CI template gradients. Terms are ordered by the weighted mean of their location along 5-percentile bins of the gradient. Source data are provided as a Source Data file. Abbreviations: CI cognitively impaired, FC functional connectivity, ROI region-of-interest, SC structural connectivity.

tau may initially follow physical channels to second-order sites, from which diverging patterns of projection can result in synchronization of regions not physically directly connected to one another. In other words, there may be initial transneuronal spread of tau manifesting along intact monosynaptic connections primarily in the temporo-posterior network along a posterior-anterior gradient. Guided by polysynaptic connections, functional (hyper-)synchrony, metabolic activity, and/or selective regional vulnerability[69], tau may then accumulate in a more widespread fashion across several regions along the

functional gradient (a gradient that is refined during postnatal development[46] and altered with aging[70] which may collectively predispose those regions to abnormal protein accumulation in certain individuals). Key to our analysis is the inclusion of structural and functional subject-specific hubs, as regions may lose their hub status with advanced disease progression[71,72]. In sum, in typical AD (encompassing the majority of our symptomatic sample), tau may initially propagate along structural (temporo-occipital) connections, followed by significant accumulation within the regions that are functionally or

metabolically co-active, thereby further amplifying the course that was initially set by structural connectome organization[73–75].

Different hypotheses may underlie the observed tau patterns. One of the leading hypotheses proposes a model of prion-like spreading based on in-vitro and rodent work[76,77]. In support of this, studies showed both retrograde and anterograde tau spread to regions with strong synaptic connections as well as tau within the tracts connecting those regions, favoring a model of structural trans-synaptic propagation[19,78]. Tau has also been shown to spread across synapses in an activity-dependent manner both in-vitro and in-vivo[79,80]. Optogenetic stimulation of tau-harboring neurons led to faster tau accumulation in connected regions[81]. However, apart from prion-like spreading, higher tau load may also arise from tau seeds with local replication at different epicenters from Braak III onwards[82]. Montal et al.[83] postulated that tau preferentially accumulates in regions with high *APOE* and glutamatergic synaptic gene expressions, supporting an alternative role for shared genetic susceptibility in focal tau accumulation[74,75]. In support of this, our overlapping axes of tau spread (captured by orthogonal gradients) may reflect that the pathology can originate from multiple epicenters (e.g., MTL and precuneus), previously termed secondary seeding regions[84]. Our work supports the concept that tau may accumulate at specific locations along the cortical hierarchy because their topological patterns show similar functional and/or microstructural properties and regional vulnerability.

In regard to brain inflammation, the spatial distribution of TSPO-PET uptake was significantly related to the principal structural and functional organization of the neocortex, with the former demonstrating a stronger association than the latter in our cohort. This pattern of inflammation, in combination with its spatial colocalization with the primary Δtau gradient, may support the idea that reactive microglia participate in tau seeding/spreading[85,86]. One biological explanation of microglia driving tau accumulation could be that microglia phagocytose the tau and therefore might play an important role in spreading of tau pathology throughout the brain[86]. A recent in-vivo human study demonstrated that brain regions with high TSPO-PET binding show increased connectivity with other high-TSPO binding regions, suggesting that microglia activation distributes along connectivity-based pathways similar to tau[87]. However, they did not directly look at spatial colocalization between TSPO and tau-PET. Herein, we show an overlap of the primary structural connectome, TSPO-PET and Δtau-PET gradients with Braak-like patterns, demonstrating a close spatial correlation between connectivity, microglia reactivity, and tau based on in-vivo human imaging data. Future research could explore whether early-stage increases in TSPO-PET signals are associated with an inflammatory state linked to amyloid removal, while later-stage signals may reflect an increased response to tissue damage which, in turn, may induce more tissue damage.

Finally, our results on the integration of the dominant connectome gradient axes with cognition demonstrated that global brain organization shapes the tau-cognition relationship. We observed consistently high tau effects on cognition for the entire cortical gradient, with progressively stronger effects towards the transmodal or anterior regions for most domains tested. While this study represents an advancement in correlating gradient-derived tau SUVR (within gradient-derived meta-ROIs) with cognition, prior work observed topology-dependent correlations between fluid intelligence and gradient-derived Aβ SUVR particularly in transmodal cortices[36]. We believe this analytical framework based on macro-scale gradients can be applied in future studies to study brain-behavior relationships in a more sensitive way by capturing domain-specific impairment through gradient-based meta-ROIs (orderly sequences in impaired subnetworks along the gradient). Furthermore, our observed interaction between tau and gradient score supports previous work on greater inter-network connectivity and network specialization blurring in AD[88–90]. This increased connectedness between different networks

may in-turn drive future protein propagation and cognitive decline[56,91]. Taken together, our findings support the contribution of connectome-related pathology distribution on cognitive impairment.

Our work has further implications in a clinical context. Based on our finding that tau in patient-specific gradient-derived hubs can predict regions of future tau accumulation, clinical trials of brain stimulation aimed at targeting such patient-tailored connector hubs may be more sensitive in slowing the disease progression. More broadly, given that brain activity dynamics may exist in a low-dimensional functional state space[40], treatments that aim to modify such pathological state (e.g., firing instability of neural circuits and impaired synaptic plasticity[92]) may be more effective than targeting a pathological substrate in a priori isolated region. Gradients provide a promising avenue to capture these overarching spatial arrangements of cortical features. In addition, sensitive methods such as electrophysiology are warranted to elucidate the role of hyperactive neurons in driving tau progression in the presence of Aβ[64,93,94]. Targeting network dysfunction or hyperexcitability has a great potential to reduce tau production and rescue cognitive impairment[95,96] and integrating this with connectome gradients is an important topic of further study. Lastly, based on our finding of a global functional gradient contraction, post-treatment evaluation of gradient normalization (e.g., increased segregation at the whole-brain level) has promise as a secondary and low-cost biomarker of trial outcome and treatment response.

This study should be interpreted considering limitations. First, we focused on the highest-variance gradients per modality. While we unveiled the brain-wide trends of connectivity and pathology similarity in individuals along the more typical AD spectrum, the first few gradients are not sufficient to capture the asymmetry and different phenotypes associated with a more heterogeneous AD cohort. Future work should map individual ΔPET maps to a larger set of connectome gradients with smaller explained variance to predict individualized (patient-specific) progression maps[97]. Second, the fMRI and dMRI-derived connectivity measures are proxies of neuronal co-activation and white matter pathways, respectively. In-vivo tractography is not sensitive enough to map the full spatial extent of existing tracts, particularly short-range connections, which may be key in tau spreading. Third, we did not acquire longitudinal data for TSPO-PET as we did for tau. Finally, the TSPO-PET uptake is not a direct measure of the inflammatory response and may reflect binding density and/or metabolic activity rather than an activation phenotype of (micro-)glia[98,99]. Moreover, certain studies have identified the presence of TSPO-negative reactive microglia[100]. Together, this suggests that the TSPO-PET signal may not capture all reactive microglia and that the lack of a TSPO-PET signal does not unequivocally signify a lack of microglial reactivity. These limitations emphasize the need for additional research to better understand the distinct cellular compartments and cell states/phenotypes giving rise to the PET signal. Nevertheless, we previously showed that TSPO-PET has been widely used as a biomarker of inflammation in diseases of the central nervous system, showing robust increases in AD compared to controls[101]. Last, it should be recognized that the brain connectome has a dual function, serving both as a direct axonal conduit for tau spread and as a dynamic influence on the progression of tau, for instance through disease-induced network dysfunction and decoupling or Aβ-induced neuronal hyperexcitability[102]. The effects of these neuroimaging-based structural and functional measures have often been modeled separately in previous in-vivo imaging studies. However, functionally connected areas are inherently structurally connected, either trans-synaptically or due to common input areas. Hence, dMRI- and fMRI-based metrics are interlinked and model overlapping processes to a certain extent. Importantly, what is commonly overlooked in interpreting dMRI and fMRI-based connectivity is that these metrics differ in their methodological sensitivity across spatiotemporal scales, likely capturing nuances of mono- vs. polysynaptic connections and local versus global

connectivity[103,104]. In addition, caution is warranted due to differential confounds introduced during data acquisition and processing of dMRI and fMRI data. Nonetheless, our dMRI/fMRI-based results match pre-clinical tau progression models at these anatomical scales.

In conclusion, our study offers insight into the system-level spatial alignment between connectome and pathological (tau/inflammation-PET) variations across the neocortex in an unbiased and data-driven manner. We showed that individual connectome gradients are altered in AD, interact with tau to alter cognition, and drive tau accumulation in a stage- and region-dependent manner, such that temporo-occipital tau accumulation is facilitated by high tau levels within gradient-derived structural hubs while frontoparietal tau accumulation is facilitated by high tau levels within gradient-derived functional hubs. The strengths of our study involve its multimodal, subject-specific nature and the application and comprehensive validation of the emerging gradient approach in AD. Our findings support the use of connectome gradients as a framework for understanding the distribution of pathological proteins along the major axes of brain organization underlying specific cognitive domains. The current work provides a model to arrange AD pathological features along macro-scale organizational axes and is a stepping stone for future studies targeting connectome-related pathology accumulation in AD and different neurodegenerative disorders.

## Methods

The McGill University, the Montreal Neurological Institute (MNI) PET working committee, and the Douglas Mental Health University Institute Research Ethics Board (Mental Health and Neuroscience sub-committee of the CIUSSS ODIM REB) provided ethical approval (IUSMD-16-60).

### Participants

The Translational Biomarkers in Aging and Dementia (TRIAD) cohort was launched in 2017 as part of the McGill University Research Center for Studies in Aging and aimed at describing biomarker trajectories and interactions as drivers of dementia. The current study included 213 participants from the TRIAD cohort, of which 103 CN A-, 35 CN A+, and 75 CI A+ with mid cognitive impairment (MCI) or AD dementia. All participants underwent [18]F-MK6240 tau-PET, [18]F-NAV4694 Aβ-PET, structural-, diffusion-, and resting-state functional MRI, *APOE*-ε4 genotyping, and cognitive testing. Ninety-four and 87 participants additionally underwent [11]C-PBR28 TSPO-PET and follow-up tau-PET (420 ± 79 days), respectively. Only high-affinity binders according to the *TSPO* rs6971 polymorphism were scanned. The demographics are outlined in Table 1. Sex was based on self-report. None of the patients presented with neuropsychiatric disorders other than MCI or AD dementia. All participants gave their written informed consent prior to inclusion in the study. They received a compensation to cover travel expenses and time.

### T1-weighted acquisition and processing

T1-weighted MRI images were acquired on a 3 T Siemens Magnetom using a volumetric magnetization prepared rapid acquisition gradient echo (MPRAGE) sequence (TR: 2300 ms, TE: 2.96 ms). Each MRI was skull stripped using our in-house ML-based tool ICVMapper[105] and further processed using FreeSurfer v7[106], including motion correction, intensity normalization, hemispheric separation, and white/pial tissue segmentation and parcellation. Incorrect segmentation of pial surfaces, which appeared particularly in the MTL of participants with substantive atrophy, was manually corrected. In addition, each preprocessed MRI was registered to the MNI template using affine and nonlinear transformation with the ANTS v2.3.1 registration toolbox. Cortical regions (nodes) were parcellated based on three different brain atlases: (i) an in-house developed high-resolution parcellation based on the multi-modal Glasser atlas re-parcellated into equally-

sized sub-regions-of-interest (ROIs) of ~512mm³ totaling 1318 nodes, (ii) the structural-based Desikan-Killiany-Tourville (DKT) atlas implemented in FreeSurfer consisting of 66 nodes, and (iii) the functional-based Schaefer atlas consisting of 100 nodes with addition of the hippocampus based on the Harvard-Oxford atlas. The parcellation described in (i) allowed for an unbiased analysis of continuous changes in both structure and function between nodes while keeping a reasonable resolution for PET and diffusion tractography. We replicated our main results across all atlases to assure robustness to the choice of parcellation scheme. Last, we overlaid our customized high-resolution brain atlas with the six Braak regions as defined in[107]; the parcels overlapping with multiple Braak regions were assigned to only one Braak region based on highest percent overlap.

### PET acquisition and processing

The [18]F-MK6240, [18]F-NAV4694, and [11]C-PBR28 PET scans were acquired using a Siemens high-resolution research tomography (HRRT) scanner. The images were acquired at 90–110, 40–70, and 60–90 min following radiotracer injection, respectively, and reconstructed using an ordered-subsets expectation maximization (OSEM) algorithm on a 4D volume with 4 (x300 seconds), 6 (x300 seconds), and 3 (x600 seconds) frames, respectively. They were corrected for dead time, decay, random and scattered coincidences, and attenuation based on a 6-min transmission scan with a rotating [137]Cs point source. The PET images were first co-registered with rigid transformation to the participant's T1w MRI (using FreeSurfer). Parcellations were inverse-transformed from template to individual T1w MRI space for regional value extraction. The final smoothing corresponded to 8 mm full-width-at-half-maximum of the Gaussian kernel. Standardized uptake value ratio (SUVR) maps were generated for each radiotracer and normalized to the inferior portion of the cerebellar gray matter. Aβ status was determined based on visual rating of the Aβ-PET images, with the final rating based on consensus of two physicians specialized in dementia imaging.

### Resting-state fMRI acquisition and processing

fMRI data were collected with single-shot full k-space multiband echo-planar imaging (EPI) and the following parameters: TR = 0.681 s, TE = 32 ms, slice thickness = 2.5 mm, number of slices = 54, flip angle = 50 degrees, number of measurements = 870, matrix size: 88 ×88, voxel size = 2.5 mm³ isotropic, and eyes open fixed on a cross. Data were preprocessed using fMRIPrep v.20.2.3, including slice timing, skull stripping, intensity normalization, and co-registration to the MNI152 space with boundary-based rigid-body and nonlinear transformations. Post-processing nuisance regressors were based on the CompCor predefined strategy as outlined in Behzadi et al.[108] and included bandpass filtering (0.01-0.08 Hz), non-steady-state volume, head motion with linear/quadratic terms and derivatives, and six components for the anatomical + temporal CompCors. This was implemented in Python through Nilearn's *fmriprep.load_confounds* function using the CompCor strategy developed by Wang et al.[109] One control subject was discarded from further fMRI analysis due to excessive motion. The rs-fMRI data were smoothed with 4 mm full-width-at-half-maximum. More details on the fMRI processing can be found at https://fmriprep.readthedocs.io/en/latest/workflows.html.

### dMRI acquisition and processing

dMRI data were collected with EPI sequence with the following parameters: TR = 3500 ms, TE = 71 ms, flip angle = 90 degrees, field of view = 232 × 232 x 162, voxel size = 2 mm³ isotropic, and 13, 48, and 60 isotropically distributed diffusion-sensitizing gradients with b-value = 0, 1000, and 2000 s/mm², respectively, as well as five b0 images. Data were preprocessed using FSL v6.0.5[110] and MRtrix3 v3.0.3[111], including correction for susceptibility distortions, motion (both between frames and within frames), gibbs ringing, and top-up and eddy currents (and

removal of the full frame if >20% of the slices within the frame are detected as outlier based on the FSL *Eddy* report). We estimated the GM-WM-CSF response functions based on the d'Hollander algorithm[112] for 40 randomly selected individuals and generated the group response functions. We then performed multi-shell multi-tissue constrained spherical deconvolution[113], intensity normalization in the log-domain, and anatomically-constrained tractography[114] with dynamic seeding from the WM and cropping at GM-WM boundary. We used 20 million streamlines with a maximum tract length of 250 mm and a fractional anisotropy cut-off of 0.06. The whole-brain streamlines were weighted by the cross-section multipliers derived from spherical deconvolution-informed filtering of the tractogram (SIFT2). The sum of weights (that is, the sum of intra-axonal cross-sectional areas of the streamlines within a certain fiber bundle) was multiplied by the proportionality constant to generate a measure of the intra-axonal cross-sectional area of the fiber bundle capacity (FBC)[115]. The FBC provides information about the capacity of the bundle to transfer information and allows to directly compare connectomes between participants.

## Cognitive assessments

Neuropsychological evaluation consisted of memory, language, processing speed, executive function, cognitive flexibility, object recognition, word reading, and the MMSE. Delayed memory was based on a composite score of the Logical Memory Test IIA (# story units), the Free and Cued Selective Reminding Test (free and cued), the Aggie Figures Visual Learning Test (hits list B), and the Rey Auditory Verbal Learning Test (hits and A7). Immediate memory was assessed using the Logical Memory Test IA (# story units) and the Free and Cued Selective Reminding Test (free and cued). Language was assessed using the Category Fluency test and Boston Naming Test. Speed was based on Trail Making Test A and WAIS-III Digit Symbol, and executive function was based on Trail Making Test B. Cognitive flexibility and word reading were assessed using the inhibition/switching and reading portions, respectively, of the D-KEFS Color-Word Interference Test. Object recognition was assessed using the BORB object decision easy B and orientation match tasks. Individual raw scores were standardized following the method described in Malek-Ahmadi et al.[116], and averaged across subdomains to create the composite scores.

## Gradient identification

A methodological overview of the data processing is visualized in Fig. 1. The steps corresponding to the gradient identification are depicted in Fig. 1b and described in detail below.

## Connectome gradients

Functional connectivity (FC) and structural connectivity (SC) matrices were generated by mapping Pearson's R correlation coefficients (for FC) and a number of reconstructed cross-section streamlines (for SC) to each of the brain parcellations (see above). These matrices were then thresholded to retain the top 20% connections per row, Fisher-transformed (for FC only), and converted into cosine similarity matrices[34]. We applied an unsupervised and non-linear dimensionality reduction technique, diffusion embedding (DE), to transform the high-dimensional connectome data into the low-dimensional embedded space. The Euclidean distance between the data points within this embedded space equals the diffusion distance between probability distributions centered at those points[42]. Unlike other dimensionality reduction techniques such as isomaps or PCA, the DE maps (i) retain the global relationships between data points, (ii) are more robust to noise[117], and (iii) are computationally fast with single-parameter optimization ($\alpha = 0.5$ and $t = 0$ to retain the global relations between data points in the embedded space) (more details on the choice of DE and $\alpha$ are outlined in the SI of Margulies et al.[34]). We extracted the principal ten gradient components for each of the connectivity matrices using BrainSpace v0.1.3[118] in Python v3.7.6. Each gradient represented an orthogonal axis of spatial (structural or functional) variation across the cortex, with the first gradient explaining most of the information in the participant's connectome data. Cortical regions (nodes) that are strongly interconnected were located closely together on the gradient, while weakly connected nodes were further apart. Using the principal gradients per modality, we created a simplified coordinate system of spatial variation in connectivity. The relative positioning of nodes within this coordinate system informs on the (dis)similarity of their brain-wide connectivity strength. Each gradient was realigned via the Procrustes algorithm to their corresponding group-wise (template) gradient derived from the group-averaged FC or SC matrix[118]. We further validated our procedures by applying different (i) sparsity thresholds (10-30%), (ii) similarity kernels (cosine; normalized angle), (iii) template alignment (cohort-level; group-level), as well as (iv) dimensionality reduction (DE; PCA). All validation results are shown in the Supplementary Data.

## PET gradients

Group-wise PET covariance matrices were generated by mapping the tau-PET and inflammation-PET data to each of the brain parcellation schemes. Similarly, a longitudinal tau-PET covariance matrix was created based on: (follow-up − baseline SUVR)/Δtime[days] thresholded at 50% sparsity. We extracted the principal ten gradient components for each of the PET covariance matrices as outlined above. We performed two sensitivity analyzes as follows. First, we generated the group-wise PET covariance matrices and corresponding gradients based on biomarker status: A-T-, A+T-, or A+T+. Amyloid status was determined as described above and tau status was based on visual rating of the tau-PET images with high retention in early Braak areas in line with Seibyl et al.[119]. Second, we applied different thresholding methods to the PET covariance matrices (20-50%). All sensitivity analyzes are shown in the Supplementary Data.

## Statistical analyses

Statistical analyses were performed in Python v3.7 and included: (i) group-wise differences in connectome gradients, (ii) group-level associations between connectome and PET gradients, (iii) subject-level associations between longitudinal tau accumulation and baseline tau within gradient-derived subject-specific hubs, and (iv) subject-level associations between connectome gradients, tau, and cognition.

The main analyses with connectome gradients are summarized in Fig. 1c. To investigate regional differences in the connectivity gradient scores between the diagnostic groups, we applied linear regression models adjusted for age, sex, and *APOE*-ε4. T-maps were adjusted for family-wise errors due to multiple comparisons with a false-positive rate at two-sided P < 0.01 and cluster-wise threshold of 500 voxels. Summary network-based spider plots were generated by entering the network-averaged gradient scores as the dependent variable into a linear regression with the diagnostic group as the independent variable and adjusted for the aforementioned covariates. Finally, we correlated our positive and negative t-statistic maps of between-group differences in gradient scores with the NeuroSynth database (https://neurosynth.org/). We generated word clouds corresponding to the regions with the highest Pearson's R correlations, after removing demographical and anatomical terms.

Next, we investigated if connectome gradients relate to pathology distribution. Per diagnostic group, correspondence of the first three gradients of connectivity ($G_{FC}$, $G_{SC}$) with the gradients of PET ($G_{TAU}$, $G_{INFLAM}$) was calculated via Spearman's rank correlation between spatially matched nodes. Similarly, we investigated the correlation of $G_{FC}$, $G_{SC}$, or $G_{INFLAM}$ with the gradients of longitudinal changes in tau-PET ($G_{\Delta TAU}$). All correlations were adjusted for spatial autocorrelation based on Variogram matching with 1000 permutations[120]. The RMSE and $R^2$ were calculated for linear and non-linear fits between the

gradients, yielding the most optimal results for a third-order polynomial (Supplementary Tables 1-2).

Next, we investigated if tau deposition within gradient-derived connectivity hubs drives tau accumulation. Specifically, we investigated the role of tau within (gradient-derived) functional and structural connectivity hubs on tau accumulation within each brain ROI per individual participant as follows (Fig. 5a): 1) Select the top most-connected 50 network hubs (based on the highest brain-wide degree of the participant's thresholded and binarized [functional or structural] connectome), 2) For each of the brain ROI, select the 10 nearest connected hubs, i.e., with smallest Euclidean distance in 3D connectome gradient space to the region, 3) Within these 10 hubs, average the tau SUVR to extract $tau_{FC\_hubs}$ and $tau_{SC\_hubs}$, and 4) Perform a linear regression across participants: $\Delta tau_{ROI} \sim baselinetau_{ROI} + tau_{FC\_hubs} + tau_{SC\_hubs} + covariates(age, sex, APOE\varepsilon4)$. The resulting T-map was adjusted for family-wise errors due to multiple comparisons with a false-positive rate at two-sided $P < 0.001$ and a cluster-wise threshold of 500 voxels, and projected onto the brain surface using trilinear interpolation. The procedure was repeated using the Schaefer atlas, by selecting the top brain-wide 15 hubs and the top 5 (gradient-based) nearest connected for each ROI; to maintain a similar proportion of hubs used relative to the total number of ROIs/parcellations. Last, we performed three sensitivity analyses to dissect the relative importance of 1) hub status, 2) connectivity to the ROI, and 3) tau within hubs. For the first sensitivity analysis, we selected the 10 regions with the nearest Euclidean distance in gradient space to the ROI irrespective of their hub status and averaged the tau SUVR within these gradient-derived connections. Then, we repeated the aforementioned linear regression analysis. For the second sensitivity analysis, we selected 10 non-hubs with the furthest Euclidean distance to the ROI and repeated the procedure. For the third sensitivity analysis, we repeated the main analysis using FC and SC gradient-based connectivity as the main predictors rather than tau within the connections: $\Delta tau_{ROI} \sim baselinetau_{ROI} + FC\_hubs + SC\_hubs + covariates(age, sex, APOE\varepsilon4)$.

Finally, we investigated if connectome gradients relate to cognitive impairment and cognitive decline. To this end, we first investigated at the cross-sectional level whether the appearance of both regional tau and gradient alterations are associated with more cognitive impairment through an interaction model adjusted for age, sex, APOE-ε4, and education. Second, at the longitudinal level, we performed a linear mixed effects model of the interaction between time (visit 1, 2 or 3), regional tau, and regional gradient score on cognitive change over 2 years, adjusted for the aforementioned covariates and random factor for subject ID; only subjects with at least one follow-up cognitive score were included. The resulting T-maps were adjusted for family-wise errors due to multiple comparisons with a false-positive rate at two-sided $P < 0.01$ and cluster-wise threshold of 500 voxels and projected onto the brain surface using trilinear interpolation. Individual gradients were aligned to the full cohort-level template. Third, we investigated whether the relationship of PET with cognition gradually changed along the $G_{FC}$ or $G_{SC}$ axis. To this end, we performed partial regression to correlate each of the cognitive composite scores with the PET SUVR averaged within six discretized (equally sized) spatial clusters along $G1_{FC}$ or $G1_{SC}$, adjusted for age, sex, APOE-ε4, education, and FDR multiple comparisons (across 9 composite tests x 6 gradient-based ROIs x 2 modalities). The resultant correlation coefficients (Fisher Z-transformed) were plotted against bin-ordering and fed into a linear regression to test for a low-dimensional representation of gradual (or progressive) tau changes with cognition along the connectome gradient. We tested the stability of the outcome through different bin sizes, including 6 and 20 bins, and calculated FDR-adjusted P-values (across 9 composite tests x 2 modalities) both across all participants and within A+ separately. Similarly, we performed

partial regression to correlate cognition with PET SUVR averaged within the six Braak stages. Finally, we performed a NeuroSynth meta-analytic decoding of each of our primary template gradients using the 22 coherent mental functions selected previously[121] from the v5_topic_50 list (https://neurosynth.org/analyzes/topics/v5-topics-50/). For analysis and plotting, we followed the same method outlined by Margulies et al.[34].

## Reporting summary
Further information on research design is available in the Nature Portfolio Reporting Summary linked to this article.

## Data availability
All requests for raw and analyzed data and materials will be promptly reviewed by McGill University to verify if the request is subject to any intellectual property or confidentiality obligations. Anonymized data will be shared upon request to the study's senior author from a qualified academic investigator for sole the purpose of replicating the procedures and results presented in this article. Any data and materials that can be shared will be released via a material transfer agreement. Data are not publicly available due to information that could compromise the privacy of research participants. Related documents, including study protocol and informed consent forms, can similarly be made available upon request. NeuroSynth database with meta-analytic topic terms is available at https://neurosynth.org/analyses/topics/v5-topics-50/. Source data are provided in this paper.

## Code availability
Source code is available on GitHub at: https://github.com/AICONSlab/AD_connectome_gradients. We also provide an example dataset necessary to replicate, interpret, and build on the findings reported in the paper.

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

## Acknowledgements

We express our gratitude to Shruti Patel for their assistance in formatting the manuscript, and to Soumik Farhan for their assistance in NeuroSynth analysis. This work was funded by the Alzheimer's Association (AARF-21-848556), the Canadian Institutes for Health Research (CIHR-PJT-178059), the Alzheimer's Society (ALZ23-05), and the Sandra Black Center for Brain Resilience and Recovery. Dr. Goubran is supported by the Canada Research Chairs program (CRC-2021-00374) and the Stuss Young Investigator Award. Dr. Ottoy is funded by the Alzheimer's Association (AARF-21-848556) and by the Canadian Vascular Training (VAST) Platform (2023-2024). Dr. Kang is funded by the CIHR-187890 and the Alzheimer's Society Research Program. The Translational Biomarker for Aging and Dementia (TRIAD) cohort is supported by the Weston Brain Institute, the Canadian Institutes for Health Research (CIHR) (MOP-11-51-31; RFN 152985, 159815, 162303), the Canadian Consortium of Neurodegeneration and Aging (CCNA; MOP-11-51-31 -team 1), the Alzheimer's Association (NIRG-12-92090, NIRP-12-259245), the Brain Canada Foundation (CFI Project 34874; 33397), the Fonds de Recherche du Québec – Santé (FRQS; Chercheur Boursier, 2020-VICO-279314), and the Colin J. Adair Charitable Foundation. Dr. Rosa-Neto and Dr. Gauthier are members of the CIHR-CCNA. Dr. Bo-yong Park is funded by the National Research Foundation of Korea (NRF-2021R1F1A1052303; NRF-2022R1A5A7033499), Institute for Information and Communications Technology Planning and Evaluation (IITP) funded by the Korea Government (MSIT) (No. 2022-0-00448, Deep Total Recall: Continual Learning for Human-Like Recall of Artificial Neural Networks; No. RS-2022-00155915, Artificial Intelligence Convergence Innovation Human Resources Development (Inha University); No. 2021-0-02068, Artificial Intelligence Innovation Hub), and Institute for Basic Science (IBS-R015-D1).

## Author contributions

Conceptualization: J.O., M.S.K., S.E.B., and M.G. Data curation: J.O., M.S.K., G.B., F.Z.L., T.A.P., N.R., J.S., J.T., and P.R.N. Formal analysis: J.O., M.S.K., J.X.T.M., L.B., and M.G. Funding acquisition: J.O., M.S.K, S.E.B., P.R.N., and M.G. Methodology: J.O., M.S.K., J.X.T.M., R.V.W., B.P., L.B., J.F.A., S.H., B.C.B., and M.G. Project administration: J.O. and M.G. Resources: B.S., J.M., J.P.S., and S.G. Software: J.O., M.S.K., R.V.W., B.C.B., and M.G. Supervision: S.E.B. and M.G. Writing, original draft: J.O. and M.G. Writing, review, and editing: all authors.

## Competing interests

The authors declare no competing interests.
