## [Peer Review File · Nature Communications]

Tau follows principal axes of functional and structural brain organization in Alzheimer's disease.REVIEWER COMMENTS

Reviewer #1 (Remarks to the Author):

I consider this exceedingly comprehensive and innovative work by Ottoy et al as a major step in advancing our understanding the pathogenesis of Alzheimer's disease. I commend the rigorous efforts the authors have put into this manuscript. Some of my comments are mainly conceptual and my aim is to genuinely engage in a discussion with the authors; they can integrate parts of this discussion in the manuscript as they see fit. Other comments (major and minor) pertain to methodological factors and have more concrete implications.

Conceptual comments

- The authors cite a leading hypothesis in the field - the spreading hypothesis - as a central tenet underlying their results and write as if it is crystal-clear that tau spreads in a trans-neuronal manner. However there is some evidence that is irreconcilable with this hypothesis. For instance, some patients exhibit focal accumulation of pathology (tau, TDP-43) that remains spatially circumscribed for many years (even decades) in some patients (e.g. <https://n.neurology.org/content/97/19/908.abstract>). Moreover, some large-scale networks are almost unvariably protected against tau accumulation even in advanced disease stages (e.g., sensorimotor). While these networks are relatively isolated, they are not disconnected from the rest of the brain. Additionally, data in humans is based on moderate correlations and the spreading mechanism itself has not been observed/proven. I interpret the results from this manuscript (and other work) in a different way. It looks like tau accumulates along gradients not because it "spreads", but because the topological patterns expressed by these gradients share structural and/or functional properties (e.g., excitatory/inhibitory balance, neurotransmission systems, genetic factors, molecular characteristics). Amyloid accumulation aligns with this idea, as it simultaneously accumulates in areas at the far-end of the principal gradient and in regions that highly overlap in their genetic expression and metabolic demand (<https://www.nature.com/articles/s41467-021-24306-2>). I feel this is a much more unifying account to explain these results than "spread", which does not explicitly take into account the factors I just cited.

- Most therapeutics approaches to Alzheimer's disease have largely focused on cellular and molecular properties of the brain and have thus far failed to provide solely needed treatments aside from Lecanemab which has at best a moderate effect on slowing cognitive decline. The authors claim that their novel approach could help optimizing clinical trials in halting or preventing tau progression. Do they think the field should explore therapeutics avenues targeting the stabilization of large-scale network homeostasis in lieu of/with therapies aimed at molecular properties? Maybe stopping the disruption of brain waves within and across gradients would prevent a vicious circle fostering the production of amyloid and tau (I acknowledge this is speculative). I often wonder about the potential (and hypothetical) outcomes of a mixed trial with interventions aimed at both the system-level physiology and molecular properties (eg <https://academic.oup.com/brain/article/145/11/3776/6701823?searchresult=1>). I wonder what the authors think about this and how their results could contribute to this question.

- How do the authors concretely see their framework being applied to clinical endeavors, either clinical practice or clinical trials? While it is innovative and extremely informative to understand the pathogenesis of Alzheimer's disease, this is a highly enriched cohort that is not representative of

what is done in clinical care routine, even in highly specialized settings. There are also several technical factors that could hamper the deployment and interpretability of such a model in clinical settings.

Major comments

- There are differential relationships between tau gradients and functional or structural gradients. Have the authors considered performing a “goodness-of-fit” analysis between tau gradients and functional/structural gradients? This would provide a very straightforward interpretation of the overlap between these modalities.
- Given that gradients are extremely similar across CI and CU (and ATN status), have the authors considered including all participants in the same analysis to generate the gradients, and subsequently assess how pre-defined groups differ on these gradients? The beauty and convenience of data-driven approaches such as DE and PCA is that they leverage the breadth of the whole dataset at hand without bias from a priori classifications. I believe it would significantly streamline the interpretation of the results in addition to providing a unifying framework for their data.
- The authors cite the caveats of different atlases and I commend that they assessed the impact of various atlases and methodological parameters on their results. I wonder why they did not consider using a whole-brain approach instead? PCA and DE are particularly great at detecting physiologically relevant patterns in high-dimensional space and providing whole-brain patterns could improve predictions.
- In the absence of an independent sample to validate their results, statistical models should at the very least include a k-fold cross-validation where applicable.
- To further improve the generalizability of their results about brain-behavior correspondances, the authors may consider using Neurosynth decoding to assess the association between their gradients and an independent, large database of functional studies reflecting a wide range of mental functions (<https://neurosynth.org/decode/> now implemented through <https://nimare.readthedocs.io/en/latest/generated/nimare.decode.discrete.NeurosynthDecoder.html>).

Minor comments

- I assume the authors mean-centered their images prior to generating the gradients (or discarded the mean image). Can they confirm this?
- Beyond using motion regressors in their fMRI analysis, what were the thresholds for xyz - pitch/yaw/roll – framewise displacement to exclude subjects? Were some participants discarded for analysis?
- Unthresholded maps would be great where applicable (with indication of FDR correction on the colorbar, e.g., Figure 5).

Nick Corriveau-Lecavalier, PhD

Reviewer #2 (Remarks to the Author):

This manuscript reports the use of a multimodal approach to MRI and PET data that decomposed structural and functional brain networks into principal gradients. PET data describing tau deposition and longitudinal change and also inflammation were similarly described in terms of gradients. There were 213 participants, about half amyloid positive that included cognitively normal and cognitively impaired (CI). A subset of 94 had TSPO inflammation PET. The gradients that were defined through these analyses generally paralleled some of the gradients reported in earlier studies using this approach to define neural systems involved in cognition. There were gradients that reflected hierarchical information processing (sensory/motor and auditory/visual) as well as gradients that seemed to be driven more by topography (superior/inferior and anterior/posterior). Extensive data analysis revealed a number of key findings: (1) Functional networks showed evidence of contraction, or de-differentiation in CI participants (2) Gradients were correlated with the distribution of both tau pathology and inflammation (3) Structural and functional connectivity had different associations with longitudinal tau spread (4) High tau along with contracted gradients together were associated with cognitive impairment.

The manuscript is interesting in applying an approach to defining patterns of connectivity that has been used both to understand cortical organization in relation to the neural systems involved in cognition, and also how these systems change in disease. In general, findings reproduce schema that reflect hierarchical information processing from primary sensory to multimodality cortex, and also have shown that the degree to which this organization deviates from a canonical form is somewhat predictive of cognitive impairment or behavioral abnormality. This study makes use of this approach and reports a number of findings that fit in with this general trend. However, the primary concern here is whether there is much that is new, and whether this approach to data analysis provides new insights. Specifically:

1. The finding that alterations in network structure/connectivity is associated with cognitive impairment is not new. It has been reported using many different approaches. There are also studies of relationships between protein aggregates, connectivity alterations and cognition that are similar to the results reported here – i.e. that connectivity and amyloid or tau together produce cognitive impairment. Thus, the new approach does not seem to have produced a novel finding.
2. The patterns of brain organization definitely correspond to patterns of tau deposition. This is indeed a fascinating finding. Yet it has been known for years – the fact that tau deposits in specific regions that involve association cortex but spare primary cortex is well known. I cannot find in this manuscript any novel insight or explanation for this fact. Similarly the posterior-anterior or temporal/frontal gradient defined by G1SC is widely known, and the temporal predominance of tau (relative to frontal) is better explained by existing studies showing temporal lobe epicenters than it is by this study showing relationships to these gradients. So all told, these gradients definitely replicate patterns of tau spread but are not novel and provide no explanation for these phenomena.
3. The finding that levels of tau within hubs predicts increases in tau is also quite similar to previous findings in a number of studies that have indicated that higher baseline levels of tau along with patterns of connectivity predict the rate of tau deposition. Surprisingly, none of the analyses modeling longitudinal tau accumulation examined the effect of amyloid on these measures.
4. The extension of this approach to neuroinflammation is problematic in 3 ways. First, what is the point? The authors invoke convincing models of tau spread derived from animal models to support the idea that patterns of connectivity drive tau deposition. Why should principal gradients drive

inflammation? Second, the authors do not investigate relationships between inflammation and tau deposition. It would seem that tau deposition might be a strong driver of where inflammation is found. Finally, a recent report (Nutmans et al, Nature Comm) indicates that TSPO expression is not a feature of inflammation in primate brain, casting great doubt on whether the PET ligands for this target accurately report neuroinflammation.

Reviewer #3 (Remarks to the Author):

Ottoy et al employed diffusion embedding to examine how tau spread and inflammation relate to structural and functional connectivity in early Alzheimer Disease. The main conclusions are that in the initial stage tau spread is mostly following structural connectivity patterns and in the later stages functional connectivity patterns. AD causes a contraction of the functional and structural gradient. The connectivity changes and the tau levels interactively determine cognition.

Understanding how tau spreads in early AD is of high relevance. Diffusion embedding is a highly interesting mathematical approach to model brain connectivity as demonstrated in prior papers. Its application to understand the alignment between tau spread, inflammation and gradients is novel. The individually based combined structural and functional connectivity models are innovative. The mathematical approach is sound, clearly explained and well-executed. The gradients obtained are in line with prior findings. The estimation of the alignment between molecular measures and the gradients is robust and reliable, based on a sufficiently large sample size. The authors compare their findings with other approaches, eg based on epicenters/hubs and on Braak staging, and show how the current approach is superior and able to integrate these prior findings. The paper is very well written and this complex topic is very clearly presented. My main question relates to the interpretation of the differential effect obtained from the functional and the structural connectivity based models.

Major comments:

1. The authors emphasize the distinction between structural and functional gradients. Is the divide between structural and functional the critical factor? It is well established that in the early phase neurofibrillary tangles spread from entorhinal to hippocampus to fusiform and inferior temporal cortex and that in the later phases they spread to much more widely distributed association zones. The structural connectivity gradient may better model the ventrotemporal pathway, the functional connectivity gradient may better model the widespread connections from the medial temporal system to the precuneus, angular gyrus and other DMN regions. It may not be so much the biological distinction between structural and functional that determines the results but rather differences in how well structural or functional data and the derived models map onto the local and global connections, mono- and polysynaptic connections. The functional and structural connectivity based models may differ in sensitivity for how well they capture connectivity at different anatomical scales. The distinction between a biological and a methodological explanation could be discussed more.

REVIEWER COMMENTS

We would like to greatly thank the reviewers for their positive and constructive feedback. We have addressed all the comments and feel that the quality of the manuscript has significantly improved. Please see our point-by-point response below (our responses are in blue). The updated text in the manuscript is highlighted in blue.

Reviewer #1 (Remarks to the Author):

I consider this exceedingly comprehensive and innovative work by Ottoy et al as a major step in advancing our understanding the pathogenesis of Alzheimer's disease. I commend the rigorous efforts the authors have put into this manuscript. Some of my comments are mainly conceptual and my aim is to genuinely engage in a discussion with the authors; they can integrate parts of this discussion in the manuscript as they see fit. Other comments (major and minor) pertain to methodological factors and have more concrete implications.

Conceptual comments

1.1 The authors cite a leading hypothesis in the field - the spreading hypothesis - as a central tenet underlying their results and write as if it is crystal-clear that tau spreads in a trans-neuronal manner. However there is some evidence that is irreconcilable with this hypothesis. For instance, some patients exhibit focal accumulation of pathology (tau, TDP-43) that remains spatially circumscribed for many years (even decades) in some patients (e.g. <https://n.neurology.org/content/97/19/908.abstract>). Moreover, some large-scale networks are almost unvariably protected against tau accumulation even in advanced disease stages (e.g., sensorimotor). While these networks are relatively isolated, they are not disconnected from the rest of the brain. Additionally, data in humans is based on moderate correlations and the spreading mechanism itself has not been observed/proven. I interpret the results from this manuscript (and other work) in a different way. It looks like tau accumulates along gradients not because it "spreads", but because the topological patterns expressed by these gradients share structural and/or functional properties (e.g., excitatory/inhibitory balance, neurotransmission systems, genetic factors, molecular characteristics). Amyloid accumulation aligns with this idea, as it simultaneously accumulates in areas at the far-end of the principal gradient and in regions that highly overlap in their genetic expression and metabolic demand (<https://www.nature.com/articles/s41467-021-24306-2>). I feel this is a much more unifying account to explain these results than "spread", which does not explicitly take into account the factors I just cited.

We thank the reviewer for this insightful perspective. We agree that 'trans-neuronal spreading' may not be the single mechanism underlying tau accumulation, and have now reworded the respective sections. In particular, as the topological patterns expressed by our gradients are known to also relate to other neural spatial trends including excitatory/inhibitory balance, neurotransmission systems, genetic factors, and molecular characteristics (as mentioned by the reviewer), we acknowledge that our findings could be interpreted more broadly in the context of regional vulnerability and developmental as well as aging-related mechanisms. We have now included these perspectives more extensively in the Intro and Discussions sections (see text below). We further discussed additional candidate mechanisms such as "secondary seeding" (whereby tau can originate across multiple epicentres, for instance, within regions with high APOE and glutamatergic synaptic gene expression). Additionally, *we now consistently refer to tau 'accumulation' instead of 'spread' when interpreting our results*; as 'accumulation' aligns more comprehensively with the broader context of regional vulnerability along these gradients, while 'spread'

inherently suggests a mechanism involving neuronal routes facilitating deposition from one region to another.

Intro:

(...) And finally, apart from these methodological limitations, it should be recognized that certain networks are almost invariably protected against tau despite their connectedness to high-tau regions (e.g. precuneus-sensorimotor circuit (Jitsuishi et al. 2023 Sci Rep)). Moreover, tau starts and accumulates in a heterogeneous manner across different regions in various individuals. In this regard, it is imperative to consider that the different networks being progressively affected by AD pathology may in fact be largely determined by underlying macro-scale axes of cortical organization in connectivity, microstructure, gene expression, and/or function.

To overcome the aforementioned limitations and shed light on whether the patterns of tau, neuroinflammation, and symptoms implicated in AD are related to the macro-scale organizational axes of the brain, we here contextualize the spatial maps of tau accumulation and inflammation in AD to an emerging approach in understanding brain organization: gradients of connectivity (Huntenburg et al. 2018 Trends Cogn. Sci).

Discussion:

*Different hypotheses may underlie the observed tau patterns. One of the leading hypotheses proposes a model of ‘prion-like’ spreading based on in-vitro and rodent work (Frost & Diamond 2010 Nat. Rev. Neurosci; Dujardin & Hyman 2019 Tau Biology). (...) Montal et al. (2022 Sci. Transl. Med) postulated that tau preferentially accumulates in regions with high APOE and glutamatergic synaptic gene expressions, supporting an alternative role for shared genetic susceptibility in focal tau accumulation (Grothe et al. 2018 Brain; Cornblath et al. 2021 Sci. Adv). In support of this, our overlapping modes of tau spread (captured by orthogonal gradients) may reflect that the pathology can originate from multiple epicenters (e.g., MTL and precuneus), previously termed ‘secondary seeding’ regions (Vogel et al. 2021 Nat Med). **Our work supports the concept that tau may accumulate at specific locations of the cortical hierarchy because their topological patterns show similar functional and/or microstructural properties and regional vulnerability.***

1.2 Most therapeutics approaches to Alzheimer’s disease have largely focused on cellular and molecular properties of the brain and have thus far failed to provide solely needed treatments aside from Lecanemab which has at best a moderate effect on slowing cognitive decline. The authors claim that their novel approach could help optimizing clinical trials in halting or preventing tau progression. Do they think the field should explore therapeutics avenues targeting the stabilization of large-scale network homeostasis in lieu of/with therapies aimed at molecular properties? Maybe stopping the disruption of brain waves within and across gradients would prevent a vicious circle fostering the production of amyloid and tau (I acknowledge this is speculative). I often wonder about the potential (and hypothetical) outcomes of a mixed trial with interventions aimed at both the system-level physiology and molecular properties (eg <https://academic.oup.com/brain/article/145/11/3776/6701823?searchresult=1>). I wonder what the authors think about this and how their results could contribute to this question.

We thank the reviewer for this interesting discussion topic. We also see great merit in therapies targeting the normalization of large-scale network function – as adjuvant or combinatorial with molecular strategies. In support of this idea, a study by our group at Sunnybrook showed that the levels of entorhinal (EC) tau and the connectivity between the EC and the HP (measured through electrophysiology recordings and phase-amplitude coupling analysis) in the TgF-344 rat model of AD

remained impaired even with significant A β -lowering treatment, explaining persistent spatial memory deficits (Morrone et al. 2020 Brain). Furthermore, HP network oscillations and theta–gamma coupling may arise well before A β overproduction (Goutagny et al. 2013 Eu J Neurosci), with A β oligomers inducing neuronal hyperexcitability in the earliest stages of AD (Busche et al. 2012 Proc. Natl. Acad. Sci.). As such, clearing A β (or even tau) alone may not be sufficient to fully elevate cognition in the presence of persistent network dysfunction. Stabilization of network homeostasis could thus be a promising avenue of treatment; however, neuromodulation investigations (e.g., TMS and DBS) in humans traditionally focused on specific anatomical (sub-)regions, thereby treating these regions as isolated entities (e.g., Koch et al. 2022 Brain; Ríos et al. 2022 Nat Comm; Senova et al. 2018 Trends in Neurosci). As such, these studies did not account for the overarching spatial arrangement of cortical features as uniquely captured by gradients underlying specific cognitive domains. The following has been added to the Discussion:

Given that brain activity dynamics may exist in a low-dimensional functional state space (Jones et al. 2022 Nat Comms), treatments that aim to modify such pathological state (e.g., firing instability of neural circuits and impaired synaptic plasticity (Frere & Slutsky 2018 Neuron) may be more effective than targeting a pathological substrate in a priori isolated region. Gradients provide a promising avenue to capture these overarching spatial arrangements of cortical features. In addition, sensitive methods such as electrophysiology are warranted to elucidate the role of hyperactive neurons in driving tau progression in the presence of A β (Toniolo et al. 2020 Int. J. Mol. Sci.; Zott et al. 2019 Science; Schoonhoven et al. 2023 Brain). Targeting network dysfunction or hyperexcitability has a great potential to reduce tau production and rescue cognitive impairment (Rodriguez et al. 2020 Plos Biol; Klink et al. 2021 BMC Psy) and integrating this with connectome gradients is an important topic of further study.

1.3 How do the authors concretely see their framework being applied to clinical endeavors, either clinical practice or clinical trials? While it is innovative and extremely informative to understand the pathogenesis of Alzheimer’s disease, this is a highly enriched cohort that is not representative of what is done in clinical care routine, even in highly specialized settings. There are also several technical factors that could hamper the deployment and interpretability of such a model in clinical settings.

Thank you for the opportunity to elaborate on the clinical impact. The following was added to the Discussion:

Our work has further implications in a clinical context. Based on our finding that tau in patient-specific gradient-derived hubs can predict regions of future tau accumulation, clinical trials of brain stimulation aimed at targeting such patient-tailored connector hubs may be more sensitive in slowing the disease progression. (...) Lastly, based on our finding of a global functional gradient contraction, post-treatment evaluation of gradient normalization (e.g., increased segregation at the whole-brain level) has promise as a novel secondary and low-cost biomarker of trial outcome and treatment response.

Major comments

1.4 There are differential relationships between tau gradients and functional or structural gradients. Have the authors considered performing a “goodness-of-fit” analysis between tau gradients and functional/structural gradients? This would provide a very straightforward interpretation of the overlap between these modalities.

We have now added GOF metrics for the analyses between PET and functional/structural gradients. We explored root mean squared error (RMSE) and R-squared. The results are listed below and added to Figures 3-4. In addition, the following was added to the Methods:

The RMSE and R^2 were calculated for linear and non-linear fits between the gradients, yielding the most optimal results for a third-order polynomial (Supplementary Tables 1-2).

Supplementary Table 1. Goodness-of-fit metrics corresponding to main Fig 3.

	tanh	cubic	linear
	RMSE, R^2	RMSE, R^2	RMSE, R^2
$G1_{FC}-G1_{TAU}$	0.0047, 0.41	0.0044, 0.50	0.0047, 0.41
$G1_{SC}-G2_{TAU}$	0.0028, 0.57	0.0029, 0.56	0.0034, 0.39
$G2_{FC}-G1_{INFLAM}$	0.2082, 0.24	0.1883, 0.38	0.2082, 0.24
$G1_{SC}-G1_{INFLAM}$	0.1801, 0.43	0.1683, 0.50	0.1801, 0.43

Supplementary Table 2. Goodness-of-fit metrics corresponding to main Fig 4.

	tanh	cubic	linear
	RMSE, R^2	RMSE, R^2	RMSE, R^2
$G1_{FC}-G2_{\Delta TAU}$	0.0420, 0.13	0.0387, 0.26	0.0420, 0.13
$G2_{FC}-G1_{\Delta TAU}$	0.1636, 0.12	0.1635, 0.12	0.1657, 0.10
$G1_{SC}-G1_{\Delta TAU}$	0.1448, 0.31	0.1464, 0.30	0.1567, 0.19

1.5 Given that gradients are extremely similar across CI and CU (and ATN status), have the authors considered including all participants in the same analysis to generate the gradients, and subsequently assess how pre-defined groups differ on these gradients? The beauty and convenience of data-driven approaches such as DE and PCA is that they leverage the breadth of the whole dataset at hand without bias from a priori classifications. I believe it would significantly streamline the interpretation of the results in addition to providing a unifying framework for their data.

Thank you for your suggestion. In our initial analysis, we performed a validation in which we combined all subjects to generate an unbiased cohort-level template as suggested, and found that the templates were very similar and had negligible effects on our results/findings. We have now included standard deviation (STDev) of the gradient maps across the full cohort and per diagnostic group, indicating intersubject variability with respect to the template. These STDev maps were stable across templates and cohorts as demonstrated below. We have now added this result as Supplementary Fig 2.

Supplementary Figure 2: inter-subject variability in the primary functional and structural gradients. Standard deviation (STDEV) maps of subjects' gradient scores relative to their template (per diagnostic group, see rows 1-3; or per the full cohort, see row 4). **a**, Functional gradient 1 showed the highest and lowest deviations in transmodal and unimodal regions, respectively. **b**, Structural gradient 1 showed the highest deviations in the medial temporal lobe. The resulting maps were stable across templates. The color bar ranges between 0 and 0.9 or 0.5 fraction, respectively, of the max stdev value.

1.6 The authors cite the caveats of different atlases and I commend that they assessed the impact of various atlases and methodological parameters on their results. I wonder why they did not consider using a whole-brain approach instead? PCA and DE are particularly great at detecting physiologically relevant patterns in high-dimensional space and providing whole-brain patterns could improve predictions.

The reviewer is correct that PCA and DE are great at detecting patterns and highlighting key variances. Here, we aimed to balance cross-modality resolution (2 different PET smoothed to final FWHM~8mm, fMRI smoothed with FWHM~4mm, DWI 2mm isotropic) with regional homogeneity and statistical stability. Therefore, we opted to create a customized high-resolution brain atlas (ROIs ~512mm³) guided by the boundaries of the multi-modal Glasser atlas. In addition, we replicated all results using two different low-res atlases with higher SNR. Furthermore, voxel-wise MRI/PET signals are not independent and suffer from spatial autocorrelation and partial voluming. For example, due to the high noise levels typically associated with TSPO-PET signals, further increasing the atlas resolution (to eventually individual voxels) will not unveil additional findings: see the example below for connectome-TSPO correlations in our early-stage CN A+ group using different atlas resolutions.

Figure. Spearman's correlations between TSPO-PET and connectome gradients in CN A+. High-resolution atlas yielded weak correlations (left panel), in contrast to Schaefer atlas (51 ROIs per hemisphere; middle panel) or 8 Schaefer networks (right panel).

1.7 In the absence of an independent sample to validate their results, statistical models should at the very least include a k-fold cross-validation where applicable.

Thank you for your suggestion. We have now performed leave-one-out cross-validation (LOOCV; in lieu of K-fold to preserve sample sizes) and reported the relative RMSE (as we are not training a model and freezing its weights to be tested on another dataset). The figures below show the relative RMSE for models assessing the interaction between baseline tau and gradients on tau accumulation and cognition, respectively; demonstrating good model performance across LOOCV. In addition, we removed the wording 'predict' everywhere in the paper and replaced it with 'associate to' / 'relate to'. Finally, the significance of gradient correlations was tested with null models using spatial autocorrelation-preserving surrogates based on variogram matching (1000 permutations); we changed our wording in the main Figure 3 from Pcorr to Padj to show that these P-values were adjusted for SA dependency (and not just indicate P-correlation).

Added to the Results for the models on tau accumulation and cognition:

Page 7:

We performed leave-one-out cross-validation to assess how well our model predictions matched the observed data (mean relative RMSE ~15%, Supplementary Fig. 19).

Page 8:

We performed leave-one-out cross-validation to assess how well the model predictions matched the observed data (mean relative RMSE ~16%, Supplementary Fig. 23).

Supplementary Figure 19. Leave-one-out cross-validation results of the association between tau within gradient-derived subject-specific hubs and longitudinal tau accumulation. The y-axis represents the relative root-mean-squared-error (Relative RMSE in %) for each of the regional analyses, indicating good model performance. **a**, High-resolution (Glasser) atlas. **b**, Low-resolution Schaefer atlas.

Supplementary Figure 23. Leave-one-out cross-validation results for the interaction between tau and G1 on cognition. **a**, Tau * G1 derived from functional connectome. **b**, Tau * G1 derived from structural connectome. The primary cognitive domains included MMSE and language. The y-axis represents the relative root-mean-squared-error (Relative RMSE in %) for each of the regional interaction analyses, indicating good model performance. Results are shown for the left hemisphere (with similar results for the right side).

1.8 To further improve the generalizability of their results about brain-behavior correspondances, the authors may consider using Neurosynth decoding to assess the association between their gradients and an independent, large database of functional studies reflecting a wide range of mental functions (<https://neurosynth.org/decode/> now implemented through <https://nimare.readthedocs.io/en/latest/generated/nimare.decode.discrete.NeurosynthDecoder.html>).

We thank the reviewer for this suggestion. We have now performed two Neurosynth analyses: 1) word clouds of cognitive terms corresponding to regions of largest group-wise differences in gradient scores, and 2) meta-analytic decoding of our primary template gradients for each modality. The following was added to the manuscript:

Results (page 4-5):

These between-group differences in $G1_{FC}$ were correlated with meta-analytic cognitive terms from perception (e.g., sensory-motor) to abstraction (e.g., theory of mind, semantic, retrieval) (Fig. 2d).

(...)

$G1_{SC}$ differences were correlated with meta-analytic cognitive terms from perception (e.g., pain, touch) to memory (e.g., episodic memory, face recognition) (Fig. 2h).

Methods (page 22):

Finally, we correlated our positive and negative *t*-statistic maps of between-group differences in gradient scores with the NeuroSynth database (<https://neurosynth.org/>). We generated word clouds corresponding to the regions with the highest Pearson's *R* correlations, after removing demographical and anatomical terms.

Figure 2. Functional and structural connectivity gradients are altered in AD. **a**, The first three functional connectivity gradients (G_{FC}) projected onto the left brain surface, extracted from the cohort-level connectome. Similar colors along the purple-yellow scale represent similar brain-wide connectivity patterns. The bar plots represent the corresponding network-specific average gradient scores. $G1_{FC}$, $G2_{FC}$, and $G3_{FC}$ explained 55, 17, and 9% of the information in functional connectome data, respectively. **b**, Coordinate system spanned by the first two gradients based on group-level functional connectomes, indicating $G1_{FC}$ contraction with unimodal and transmodal (DMN) regions moving closer to each other in CI participants. **c**, Histogram of gradient scores reflected global $G1_{FC}$ contraction with an expansion of scores centered around zero. **d**, Between-group comparisons (green: CN A+ vs controls; orange: CI vs controls) of network-based $G1_{FC}$ alterations (asterisks represent significant *t*-statistics at the network-level with $P < 0.05$, adjusted for age, sex, and APOE- $\epsilon 4$) using a group-level (left) or cohort-level (right) gradient realignment strategy. Word clouds of NeuroSynth cognitive terms associated with regions with positive (red) or negative (blue) *t*-statistic $G1_{FC}$ differences between diagnostic groups (using cohort-level realignment strategy). **e**, $G1_{SC}$, $G2_{SC}$, and $G3_{SC}$ explained 28, 20, and 9% of the information in structural connectome data, respectively. **f**, Coordinate system spanned by the first two gradients. **g**, Histogram of $G1_{SC}$. **h**, Between-group comparisons of network-based $G1_{SC}$ alterations and corresponding word clouds. Results are displayed for the left hemisphere; Supplementary Fig. 1, 3, and 7 show right hemisphere projections and group differences.

Results (page 9):

Last, we performed a NeuroSynth meta-analytic decoding of each of our primary template gradients (Supplementary Fig. 27), showing how cortical organization underlies cognitive functions. We observed that cognitive domains are associated with the gradient organization from the unimodal (e.g., motor, sensory perception) to the transmodal (e.g., social, negative emotion, moral, memory) poles of both the

functional and tau-PET primary gradients (Fig. 7c). While, cognitive terms of facial recognition, sensory perception, and memory are also organized at the posterior poles of both the structural, inflammation and Δ tau-PET gradients (Supplementary Fig. 27). Taken together, our results suggest that connectivity (gradient)-derived meta-ROIs represent a sensitive method to capture brain-behavior relationships.

Methods (page 23):

Last, we performed a NeuroSynth meta-analytic decoding of each of our primary template gradients using the 22 coherent mental functions selected previously (Corriveau-Lecavalier et al. 2023 Brain) from the v5_topic_50 list (<https://neurosynth.org/analyses/topics/v5-topics-50/>). For analysis and plotting, we followed the same method outlined by Margulies et al. (2016 PNAS).

a Functional gradient 1

b Structural gradient 1

c PET gradient 1

Supplementary Figure 27. Associations between meta-analytic cognitive terms and primary gradients. Cognitive terms are derived from the NeuroSynth database and ordered by the weighted mean of their location along 5-percentile bins of the primary gradients (a: functional gradient, b: structural gradient, c: PET gradients in CI) (see Margulies et al. 2016 PNAS).

Figure 7. Gradient-derived ROIs capture brain-behavior relationships. *a*, Cognitive correlates (absolute Pearson's R) of tau SUVR within $G1_{FC}$ -derived (left), $G1_{SC}$ -derived (middle), or Braak meta-ROIs (right). *b*, The resultant correlation coefficients changed in a topographic-specific manner along the $G1_{FC}$ and $G1_{SC}$ but not the Braak axes (based on linear regression between gradient bin ordering and the tau-cognition [z-scored] correlation coefficient within each bin, $P_{FDR} < 0.05$). These analyses (panels a-b) were performed in A+ and adjusted for age, sex, education, and APOE- $\epsilon 4$, and multiple comparisons. *c*, Z-statistic maps of the associations between meta-analytic cognitive terms and primary functional (left), structural (middle), and tau-PET (right) gradients. Terms are ordered by the weighted mean of their location along 5-percentile bins of the primary gradients.

Minor comments

1.9 I assume the authors mean-centered their images prior to generating the gradients (or discarded the mean image). Can they confirm this?

Yes, the fMRI images were mean-centered during post-processing by setting `demean=True` in the `load_confounds` function from `nilearn.interfaces.fmriprep`. Similarly, the dMRI images were intensity normalized using the `mtnormalise` function implemented in Mtrix after FOD generation. In addition,

we applied inter-subject connectivity density normalization of the structural streamlines (also referred to as fiber capacity bundle, see Smith et al. 2022 Aperture Neuro HBM)

1.10 Beyond using motion regressors in their fMRI analysis, what were the thresholds for xyz - pitch/yaw/roll – framewise displacement to exclude subjects? Were some participants discarded for analysis?

fMRI: we implemented the fmripred load confounds strategy by Nilearn, see Table 1 in Hao-Ting Wang et al. 2023 (and based on advice from the function developer). A parameter called *strategy* is used to pass a list of different noise regressors to include in the confounds: motion (original parameters + 1st temporal derivatives + quadratic terms + power2d derivatives), compcor (anatomical and temporal with 6 components), high pass filtering, and non-steady-state. We discarded one subject for fMRI analysis. This has now been clarified in the Methods section.

DWI: We applied a slice-to-volume motion correction strategy (motion correction within the frame using FSL *Eddy*) as well as removed the full outlier frame if >20% of the slices within the frame are detected as outliers based on the Eddy report. This has now been added to the Methods section.

1.11 Unthresholded maps would be great where applicable (with indication of FDR correction on the colorbar, e.g., Figure 5).

We appreciate this suggestion. We have added unthresholded maps to Supplementary Figs 5 and 9 (below) representing the t-stat differences in gradient scores between diagnostic groups. These unthresholded maps helped us substantially in detecting overall trends, e.g. the unimodal vs transmodal contraction of $G1_{FC}$ in CI patients compared to controls (see below Supplementary Fig. 5c, blue vs pink colors).

Supplementary Figure 5. Group-wise differences in functional gradient scores using an unbiased cohort-level realignment strategy. Panels a-b: Spider plots of the group-wise t -statistic differences in network-based gradient scores derived from the individual functional connectomes, using functional networks (Schaefer – panel a) or structural networks (Glasser – panel b), respectively. Green and orange lines indicate differences of CN A- with CN A+ and CI groups, respectively. The direction of the positive and negative t -statistic on the gradient is indicated at the bottom of the figure (e.g., regions with a negative t -stat on gradient 1 moved closer to the transmodal cortex in disease compared to controls). T -statistics were adjusted for age, sex, and APOE- $\epsilon 4$. Panels c-d: Group-wise t -statistic differences in gradient scores derived from the individual functional connectomes. Panel c and d show the differences of CN A- with CI and CN A+ groups, respectively. The direction of the positive and negative t -statistic on the gradient is indicated at the right side of the figure. T -statistics were adjusted for age, sex, and APOE- $\epsilon 4$. The thresholded maps are based on family-wise errors due to multiple comparisons with a false-positive rate at <0.01 and cluster-wise threshold of 500 voxels.

Supplementary Figure 9. Group-wise differences in structural gradient scores using an unbiased cohort-level realignment strategy. Panels a-b: Spider plots of the group-wise t -statistic differences in network-based gradient scores derived from the individual structural connectomes, using functional (Schaefer – panel a) or structural (Glasser – panel b) networks, respectively. Green and orange lines indicate differences of CN A- with CN A+ and CI A+ groups, respectively. The direction of the positive and negative t -statistic on the gradient is indicated at the bottom of the figure. T -statistics were adjusted for age, sex, and APOE- $\epsilon 4$. Panels c-d: Group-wise t -statistic differences in gradient scores derived from the individual structural connectomes. Panel c and d show the differences of CN A- with CI A+ and CN A+ groups, respectively. The direction of the positive and negative t -statistic on the gradient is indicated at the right side of the figure. T -statistics were adjusted for age, sex, and APOE- $\epsilon 4$. The thresholded maps are based on family-wise errors due to multiple comparisons with a false-positive rate at <0.01 and cluster-wise threshold of 500 voxels.

We also added unthresholded maps for the main Figures 5 and 6. Please refer to Supplementary Fig. 22 (below).

a $\Delta\tau \sim$ baseline tau in subject-specific gradient-derived hubs

b baseline cognition \sim tau * $G1_{FC}$ in A+

c baseline cognition \sim tau * $G1_{SC}$ in A+

Supplementary Figure 22. Unthresholded t-stat maps to observe overall trends. a, Tau accumulation over time within an ROI is predicted by baseline tau levels within gradient-derived functional hubs to the ROI (first column), as well as by baseline tau levels within gradient-derived structural hubs to the ROI (middle column), and by baseline ROI tau (right column). b, The interaction effect between regional tau SUVR and functional gradient score on cognition. A negative interaction effect (blue t-statistic; transmodal regions) indicates that higher tau together with higher $G1_{FC}$ score (i.e., gradient contraction towards unimodal) results in lower cognitive performance (language and MMSE scores). Similarly, a positive interaction (red t-statistic; unimodal regions) indicates that higher tau together with contracted $G1_{FC}$ towards the transmodal end is associated with lower cognitive performance. c, The interaction between regional tau SUVR and structural gradient score on cognition.

Nick Corriveau-Lecavalier, PhD

Reviewer #2 (Remarks to the Author):

This manuscript reports the use of a multimodal approach to MRI and PET data that decomposed structural and functional brain networks into principal gradients. PET data describing tau deposition and longitudinal change and also inflammation were similarly described in terms of gradients. There were 213 participants, about half amyloid positive that included cognitively normal and cognitively impaired (CI). A subset of 94 had TSPO inflammation PET. The gradients that were defined through these analyses generally paralleled some of the gradients reported in earlier studies using this approach to define neural systems involved in cognition. There were gradients that reflected hierarchical information processing (sensory/motor and auditory/visual) as well as gradients that seemed to be

driven more by topography (superior/inferior and anterior/posterior). Extensive data analysis revealed a number of key findings: (1) Functional networks showed evidence of contraction, or de-differentiation in CI participants (2) Gradients were correlated with the distribution of both tau pathology and inflammation (3) Structural and functional connectivity had different associations with longitudinal tau spread (4) High tau along with contracted gradients together were associated with cognitive impairment.

The manuscript is interesting in applying an approach to defining patterns of connectivity that has been used both to understand cortical organization in relation to the neural systems involved in cognition, and also how these systems change in disease. In general, findings reproduce schema that reflect hierarchical information processing from primary sensory to multimodality cortex, and also have shown that the degree to which this organization deviates from a canonical form is somewhat predictive of cognitive impairment or behavioral abnormality. This study makes use of this approach and reports a number of findings that fit in with this general trend. However, the primary concern here is whether there is much that is new, and whether this approach to data analysis provides new insights. Specifically:

We thank the reviewer for their helpful comments. We have further highlighted the novelty and key findings of our manuscript on 3 levels (point-by-point discussed in the responses below). Briefly, we found that:

- Q2.1 Reduced internetwork segregation and tau interact to reduce cognitive performance.
 - We have now added a longitudinal experiment to boost novelty and further strengthen this finding.
- Q2.2 Connectome gradients align with tau-PET patterns in a stage-dependent manner, such that the P-A structural gradient is better aligned with tau in early stages and its accumulation over time, while the functional gradient is better aligned with baseline tau in symptomatic stages.
 - We have now added a novel NeuroSynth analysis and discussed our points in the context of neurodevelopmental/age-related changes and regional vulnerability.
- Q2.3 Through the use of *individualized* gradient-derived network hubs of *both* dMRI and fMRI data, we show that the spatial location of a hub region within the structural and functional gradient strongly impacts its capacity to steer pathology to its neighbors.
 - As per the recent Perspective on connectivity-based modeling of neurodegenerative disease (Vogel et al. 2023 Nat. Rev. Neurosci), one of the recommended future models was proposed to include “*individualized networks based on the DWI or resting state fMRI (or both) data of a person may help to resolve differences in spread patterns.*” We have now better highlighted these novel aspects of our work compared to existing studies on tau prediction along the connectome.

2.1 The finding that alterations in network structure/connectivity is associated with cognitive impairment is not new. It has been reported using many different approaches. There are also studies of relationships between protein aggregates, connectivity alterations and cognition that are similar to the results reported here – i.e. that connectivity and amyloid or tau together produce cognitive impairment. Thus, the new approach does not seem to have produced a novel finding.

Alterations in network structure/connectivity (“network dysfunction”) can be quantified broadly. We herein employed a unique approach to demonstrate that regions exhibiting diminished inter-network differentiation (gradient contraction) and concurrently exhibiting higher tau load are associated with lower cognitive performance (tested on 9 distinct cognitive domains). The effects were globally stronger for functional compared to structural gradient contraction. The effects were also specific to certain cognitive domains (see Supplementary Fig 24). *To our knowledge, this network segregation-tau*

interaction on cognition (in a topographic-dependent manner) is novel and has not been previously reported in the literature. We further demonstrated for the first time that gradient-derived ROIs more sensitively captured this tau-cognition relationship compared to typical Braak regions. Our findings support the idea of normalizing large-scale network de-differentiation as an adjuvant or combinatorial treatment strategy to anti-amyloid/tau trials (see reviewer #1 Q1.2). Our study, however, did not investigate which one comes first (gradient contraction or tau), which is the topic of our future work. The clinical implications of these findings are summarized in Q1.3 above.

To further boost novelty and validate our findings, we have now added longitudinal cognitive data to the revised manuscript. Previous work studied the interactions of connectivity and amyloid on cognition (Van Hooren et al. 2018 *Alzheimers Res. Ther.*), or the effects of either structural or functional connectivity within certain key networks and tau on cognition (Chen et al. 2020 *eLife*; Dautricourt et al. 2021 *Ann Neurol*) and without considering longitudinal cognitive function (Boyle et al. 2023 *Alz Dem*; Mijalkov et al. 2023 *Alzheimers Res. Ther.*; Berron et al. 2021 *Brain*; Pelkmans et al. 2021 *Alzheimers Res. Ther.*; King-Robson et al. 2021 *J. Alzheimers Dis*). As such, to our knowledge, this is the first investigation of the interactions between *whole-brain functional+structural* connectivity (and gradients) with tau-PET on *longitudinal* cognitive decline in AD. Specifically, we performed a linear mixed effect model of the interaction between time, tau and gradient score on cognitive change over 2 years. Our findings strengthened our baseline results, indicating that contracted gradient scores (particularly in transmodal regions) together with higher tau levels were associated with more cognitive decline. Importantly, this relationship was considerably stronger for the structural gradient in the longitudinal compared to the cross-sectional analysis. We added this new experiment to the Methods and Results sections (page 8 and revised Figure 6, copied below). We also added new scatterplots to the original panels a-b to highlight gradient contraction effects.

Figure 6. Interaction between connectome gradient and tau on cognition. *a*, The interaction effect between regional tau SUVR and functional gradient score (aligned to the full cohort-level template) on baseline cognition. A negative interaction effect (blue t-statistic in transmodal regions) indicates that higher tau together with higher $G1_{FC}$ score (i.e., gradient contraction towards unimodal) results in lower cognitive performance (language and MMSE scores). Similarly, a positive interaction (red t-statistic in unimodal regions) indicates that higher tau together with lower $G1_{FC}$ score (i.e., gradient contraction towards transmodal) is associated with lower cognitive performance. *b*, The interaction effect between regional tau SUVR and structural gradient score on cognition. Sample sizes of A+ (panels a-b) varied across composite scores: MMSE $n=107$ and language $n=90$. *c,d* The interaction effect between time, regional tau SUVR, and functional (panel c) or structural (panel d) gradient score on 2-year cognitive change. Sample sizes of all subjects varied across composite scores: MMSE: $n_{visit1}=104$, $n_{visit2}=89$, $n_{visit3}=58$ and language: $n_{visit1}=99$, $n_{visit2}=78$, $n_{visit3}=58$. Limbic regions are blue for G_{FC} and red for G_{SC} because they are largely located on the negative vs. the more positive pole of the respective gradients; while, the prefrontal cortex (blue) is located on both the negative pole of the respective gradients.

2.2 The patterns of brain organization definitely correspond to patterns of tau deposition. This is indeed a fascinating finding. Yet it has been known for years – the fact that tau deposits in specific regions that involve association cortex but spare primary cortex is well known. I cannot find in this manuscript any novel insight or explanation for this fact. Similarly the posterior-anterior or temporal/frontal gradient defined by $G1_{SC}$ is widely known, and the temporal predominance of tau (relative to frontal) is better explained by existing studies showing temporal lobe epicenters than it is by this study showing relationships to these gradients. So all told, these gradients definitely replicate patterns of tau spread but are not novel and provide no explanation for these phenomena.

In terms of novelty, our study presents a unified framework that contextualizes AD-specific pathological changes into the larger landscape of organizational neural axes that are highlighted by the emerging literature on gradient mapping in neuropsychiatric and neurological disorders. Reviewer #1

provided an interesting perspective on how gradients may explain patterns of tau, which we have added to the Intro and Discussion. Please refer to our response to Q1.1.

We like to further highlight the 3 key novel approaches of our work. First, we modeled *subject-specific measures of both functional and structural connectomes in the same patients* across the AD spectrum (we discuss the impact and importance of this in the next question 2.3). Second, we extracted structural connectivity *gradients from dMRI* data which has not been reported across disease stages in AD (to the best of our knowledge). Third, we modeled the *longitudinal* effects of network hubs (within the structural and functional gradient space) on pathology propagation to their neighbors. We show that distance in gradient space is a strong predictor of tau accumulation. Our analysis suggests that tau may not be simply distributed along either functional or axonal connections, but instead follows connectivity gradients in a stage- and spatial-dependent manner. We have now provided potential explanations for these findings in the revised Discussion (in addition to Q1.1):

As tauopathy is thought to typically start in anatomically defined early-Braak areas, the tau may initially follow physical channels to second-order sites, from which diverging patterns of projection can result in synchronization of regions not physically directly connected to one another. In other words, there may be initial transneuronal spread of tau manifesting along intact monosynaptic connections primarily in the temporo-posterior network along a posterior-anterior gradient. Guided by polysynaptic connections, functional synchrony, metabolic activity, and/or selective regional vulnerability⁷¹, tau may then accumulate in a more widespread fashion across several regions along the functional gradient (a gradient that is refined during postnatal development⁴⁸ and altered with aging⁷² which may collectively predispose those regions to abnormal protein accumulation in certain individuals).

Finally, in line with the original gradient paper by Margulies et al. (2016 PNAS), we now added NeuroSynth analyses to the revised Figures 2 & 7 (and Supplementary Fig. 27). This analysis allowed us to assess which cognitive terms are most closely aligned with (i) group-wise differences in primary gradient scores and (ii) primary gradient meta-ROIs, allowing for an interpretation of brain-behavior relationships based on a large external database of functional neuroimaging studies. Please refer to our response to Q1.8.

2.3a The finding that levels of tau within hubs predicts increases in tau is also quite similar to previous findings in a number of studies that have indicated that higher baseline levels of tau along with patterns of connectivity predict the rate of tau deposition.

Thank you for your comment and suggestion. The reviewer is correct that prior work proposed baseline tau-PET as a proxy to predict longitudinal tau accumulation along the structural or functional connectome (Vogel et al. 2020 Nat Comms; Franzmeier et al. 2020 Sci Adv; Raj et al. 2021 Brain Connect; Yang et al. 2021 Neuroimage; Montal et al. 2022 Sci. Transl. Med.). However, none of these approaches considered that brain networks are embedded within larger gradients of hierarchical organization, and they were typically limited by using connectivity data and network nodes derived from (normative) group-based connectivity templates or a cortical parcellation atlas. As per the recent Perspective on connectivity-based modeling of neurodegenerative disease (Vogel et al. 2023 Nat Rev Neurosci), one of the recommended future models was proposed as follows: “*individualized networks based on the DWI or resting state fMRI (or both) data of a person may help to resolve differences in spread patterns.*”

Here, we used *subject-specific* connectivity data and network nodes derived from individual connectomes. We not only used subject-specific connectivity to define the functional and structural hubs, but also the *gradient-based* connectivity between brain regions and these major hubs. This is important because: 1) Subject-specific hubs account for changes in hub status with progressive disease stage; 2) Subject-specific connectivity accounts for heterogeneous changes in connectivity between regions (with tau) and hubs across subjects and disease stages; and 3) Gradients represent the *non-linear, unbiased* embedding of a region within the overarching cortical hierarchy (in contrast to independent region-to-region connectivity).

Our results reveal novel findings not reported with previous approaches: the location of hub regions within the structural or functional gradient strongly impacts the steering of pathology to lower-tau neighbors. Specifically, when a *frontoparietal* low-tau region is close to a high-tau region on the *functional* gradient it will accumulate more tau (see Fig. 5 panel C). While, when a *temporo-occipital* low-tau region is located close to a high-tau region on the *structural* gradient it will accumulate more tau.

2.3b Surprisingly, none of the analyses modeling longitudinal tau accumulation examined the effect of amyloid on these measures.

We have now included additional experiments modeling the effects of amyloid (results shown across all subjects); demonstrating that adding $A\beta$ in our models did not significantly alter our results.

1) Region-wise adjustment for $A\beta$ SUVR (see new Supplementary Fig. 21 copied below):

“Adding baseline $A\beta$ SUVR as a covariate did not significantly alter the results (Supplementary Fig. 21).”

Supplementary Figure 21. Adjustment for amyloid. Tau accumulation over time within an ROI is predicted by baseline tau levels within gradient-derived functional hubs to the ROI (first column), as well as by baseline tau levels within gradient-derived structural hubs to the ROI (second column), and by baseline ROI tau (third column) and amyloid (last column). A positive *t*-statistic (red) within an ROI indicated a positive relationship between higher tau accumulation within the ROI and higher baseline tau or amyloid within the ROI’s subject-specific hubs or ROI.

2) $\Delta\text{Tau in ROI} \sim A\beta \text{ in FC hubs} + A\beta \text{ in SC hubs} + \text{baseline } A\beta \text{ in ROI} + \text{covariates}$ (see Figure below):

Figure. Amyloid within gradient-derived subject-specific hubs predicting tau accumulation. Tau accumulation over time within an ROI is predicted by baseline amyloid levels within gradient-derived functional hubs to the ROI (first column), as well as by baseline amyloid levels within gradient-derived structural hubs to the ROI (middle column), and by baseline ROI amyloid (right column) across all subjects. A positive t-statistic (red) within an ROI indicated a positive relationship between higher tau accumulation within the ROI and higher baseline amyloid within the ROI's subject-specific hubs. Results are shown for the Schaefer atlas.

2.4 The extension of this approach to neuroinflammation is problematic in 3 ways. First, what is the point? The authors invoke convincing models of tau spread derived from animal models to support the idea that patterns of connectivity drive tau deposition. Why should principal gradients drive inflammation? Second, the authors do not investigate relationships between inflammation and tau deposition. It would seem that tau deposition might be a strong driver of where inflammation is found. Finally, a recent report (Nutma et al, Nature Comm) indicates that TSPO expression is not a feature of inflammation in primate brain, casting great doubt on whether the PET ligands for this target accurately report neuroinflammation.

We have answered the reviewer's 3 questions in order below:

1) Since previous *in-vivo* human studies demonstrated that brain connectivity is spatially associated with patterns of TSPO PET (Rauchmann et al. 2023 Neurology) and that the spread of tau is accompanied by patterns of microglia activation (shown by our collaborators Pascoal et al. 2021 Nat Med), we wanted to test if we can confirm a spatial colocalization between connectivity, microglia reactivity and tau in our cohort using subject-specific diffusion & fMRI data and brain-wide network organization. Using gradient analyses, we observed that the posterior-anterior structural gradient is primarily associated with TSPO PET (spearman rho=0.72) and tau progression (spearman rho=0.46). Furthermore, we conducted an additional experiment (see point 2 below) showing that TSPO PET is also associated with tau progression (spearman rho=0.60). The following was added to the Discussion:

*This pattern of inflammation, in combination with its spatial colocalization with the primary Δ tau gradient, may support the idea that reactive microglia participate in tau seeding/spreading (Maphis et al. 2015 Brain; Hopp et al. 2018 J Neuroinflam). One biological explanation of microglia driving tau accumulation could be that microglia phagocytose the tau and therefore might play an important role in spreading of tau pathology throughout the brain (Hopp et al. 2018 J Neuroinflam). A recent *in-vivo* human study demonstrated that brain regions with high TSPO-PET binding show increased connectivity with other high-TSPO binding regions, suggesting that microglia activation distributes along connectivity-based pathways similar to tau (Rauchmann et al. 2023 Neurology). However, they did not directly look at spatial colocalization between TSPO and tau-PET. Herein, we show an overlap of the primary structural connectome, TSPO-PET, and Δ tau-PET gradients with Braak-like patterns, confirming a close spatial correlation between connectivity, microglia reactivity, and tau based on *in**

vivo human imaging data. Future research could explore whether early-stage increases in TSPO-PET signal are associated with an inflammatory state linked to amyloid removal, while later-stage signals may reflect an increased response to tissue damage which, in turn, may induce more tissue damage.

2) We appreciate the suggestion to investigate the relationship between tau and inflammation gradients. Indeed, as anticipated by the reviewer, baseline inflammation was strongly associated with longitudinal tau accumulation (primary gradients). We think these results are of great interest to the field, and we have now added them to the main Figure 4 d & e:

Results:

$G_{\Delta\text{TAU}}$ overlapped with the Braak stages (Fig. 4b) and was aligned with G_{SC} ($\rho=0.46$, $P_{\text{adj}}<0.001$; Fig. 4c) and G_{INFLAM} ($\rho=0.60$, $P_{\text{adj}}<0.001$; Fig. 4d-e).

Figure 4. Longitudinal tau-PET gradients align with connectome gradients. *a*, The first three gradients of longitudinal tau accumulation ($G_{\Delta\text{TAU}}$) in A+, projected onto the left brain surface. *b*, Network-specific average values for $G_{\Delta\text{TAU}}$ (top) and $G_{2\Delta\text{TAU}}$ (bottom), explaining 51 and 13% of the information respectively in $\Delta\text{tau-PET}$ data, with the first gradient largely overlapping with Braak stages. *c*, Heatmap showing Spearman's rank correlation between G_{FC} or G_{SC} and $G_{\Delta\text{TAU}}$, indicating good alignment ($G_{\text{FC}}-G_{\Delta\text{TAU}}$: $\text{RMSE}=0.04$, $R^2=0.26$; $G_{2\text{FC}}-G_{\Delta\text{TAU}}$: $\text{RMSE}=0.16$, $R^2=0.12$; $G_{\text{SC}}-G_{\Delta\text{TAU}}$: $\text{RMSE}=0.15$, $R^2=0.30$). Results are displayed for the left hemisphere; Supplementary Fig. 15 shows right hemisphere correlations. *d*, Spearman's rank correlations between $G_{\Delta\text{TAU}}$ and G_{INFLAM} based on template gradients across the cohort. *e*, Spearman's rank correlations between $G_{\Delta\text{TAU}}$ and G_{INFLAM} based on template gradients in A+.

3) We have now expanded the discussion of the limitations of TSPO PET in more detail and referenced the paper mentioned by the reviewer. The following was added to the Discussion:

Finally, the TSPO-PET uptake is not a direct measure of the inflammatory response and may reflect binding density and/or metabolic activity rather than an activation phenotype of (micro-)glia (Nutma et al. 2022 Nat Comms; Owen et al. 2017 JCBFM). Moreover, certain studies have identified the presence of TSPO-negative reactive microglia (Wright et al. 2020 Front. Neurol.). Together, this suggests that the TSPO-PET signal may not capture all reactive microglia and that the lack of a TSPO PET signal does not unequivocally signify a lack of microglial reactivity. These limitations emphasize the need for additional research to better understand the distinct cellular compartments and cell states/phenotypes giving rise to the PET signal. Nevertheless, we previously showed that TSPO-PET has been widely used as a biomarker of inflammation in diseases of the central nervous system, showing robust increases in AD compared to controls⁸⁹.

Reviewer #3 (Remarks to the Author):

Ottoy et al employed diffusion embedding to examine how tau spread and inflammation relate to structural and functional connectivity in early Alzheimer Disease. The main conclusions are that in the initial stage tau spread is mostly following structural connectivity patterns and in the later stages functional connectivity patterns. AD causes a contraction of the functional and structural gradient. The connectivity changes and the tau levels interactively determine cognition.

Understanding how tau spreads in early AD is of high relevance. Diffusion embedding is a highly interesting mathematical approach to model brain connectivity as demonstrated in prior papers. Its application to understand the alignment between tau spread, inflammation and gradients is novel. The individually based combined structural and functional connectivity models are innovative. The mathematical approach is sound, clearly explained and well-executed. The gradients obtained are in line with prior findings. The estimation of the alignment between molecular measures and the gradients is robust and reliable, based on a sufficiently large sample size. The authors compare their findings with other approaches, eg based on epicenters/hubs and on Braak staging, and show how the current approach is superior and able to integrate these prior findings. The paper is very well written and this complex topic is very clearly presented. My main question relates to the interpretation of the differential effect obtained from the functional and the structural connectivity based models.

We thank the reviewer for their positive feedback, highlighting that our approach and findings are novel, well-executed, and clearly articulated.

Major comments:

3.1 The authors emphasize the distinction between structural and functional gradients. Is the divide between structural and functional the critical factor? It is well established that in the early phase neurofibrillary tangles spread from entorhinal to hippocampus to fusiform and inferior temporal cortex and that in the later phases they spread to much more widely distributed association zones. The structural connectivity gradient may better model the ventrotemporal pathway, the functional connectivity gradient may better model the widespread connections from the medial temporal system to the precuneus, angular gyrus and other DMN regions. It may not be so much the biological distinction between structural and functional that determines the results but rather differences in how well structural or functional data and the derived models map onto the local and global connections, mono- and polysynaptic connections. The functional and structural connectivity based models may differ in sensitivity for how well they capture connectivity at different anatomical scales. The distinction between a biological and a methodological explanation could be discussed more.

Thank you for this thought-provoking comment. We agree with the reviewer that methodological distinctions between *in-vivo* functional and structural connectivity (gradients) may affect results, warranting both biological and methodological explanations. The following has now been added to the Discussion:

And last, it should be recognized that the brain connectome has a dual function, serving both as a direct axonal conduit for tau spread and as a dynamic influence on the progression of tau, for instance through disease-induced network dysfunction and decoupling or A β -induced neuronal hyperexcitability (Vogel et al. 2023 Nat. Rev. Neurosci). The effects of these neuroimaging-based structural and functional measures have often been modeled separately in previous in-vivo imaging studies. However, functionally connected areas are inherently structurally connected, either trans-synaptically or due to common input areas. Hence, dMRI- and fMRI-based metrics are interlinked and model overlapping processes to a certain extent. Importantly, what is commonly overlooked in interpreting dMRI and fMRI-based connectivity, is that these metrics differ in their methodological sensitivity across spatiotemporal scales, likely capturing nuances of mono- vs. polysynaptic connections and local versus global connectivity (Benkarim et al. 2022 Neuroimage; Avena-Koenigsberger et al. 2017 Nat Rev Neurosci). In addition, caution is warranted due to differential confounds introduced during data acquisition and processing of dMRI and fMRI data. Nonetheless, our dMRI/fMRI-based results match pre-clinical tau progression models at these anatomical scales.

REVIEWERS' COMMENTS

Reviewer #1 (Remarks to the Author):

The authors addressed all my comments with impressive rigor. I have no further comments or concerns.

Reviewer #2 (Remarks to the Author):

Thank you for the responses

Reviewer #3 (Remarks to the Author):

My comments have been addressed satisfactorily. The application of diffusion embedding modeling in AD is innovative. The unified mathematical framework is very suitable for addressing the research questions and yielded novel insight into the spatial alignment between structural, functional, inflammatory and tau related gradients.